# Multiomics analyses reveal dynamic bioenergetic pathways and functional remodeling of the heart during intermittent fasting

**Thiruma V Arumugam[1,2,3]\*[†], Asfa Alli-Shaik[4][†], Elisa A Liehn[5,6,7], Sharmelee Selvaraji[2,8], Luting Poh[2], Vismitha Rajeev[2], Yoonsuk Cho[3], Yongeun Cho[3], Jongho Kim[3], Joonki Kim[2,9], Hannah LF Swa[4], David Tan Zhi Hao[2], Chutima Rattanasopa[10,11], David Yang-Wei Fann[2], David Castano Mayan[10], Gavin Yong-Quan Ng[2], Sang-Ha Baik[2], Karthik Mallilankaraman[2], Mathias Gelderblom[12], Grant R Drummond[1], Christopher G Sobey[1], Brian K Kennedy[2,13], Roshni R Singaraja[14], Mark P Mattson[15], Dong-Gyu Jo[3], Jayantha Gunaratne[4,16]\***

**\*For correspondence:**
g.arumugam@latrobe.edu.au
(TVA);
jayanthag@imcb.a-star.edu.sg
(JG)

[†]These authors contributed
equally to this work

[1]Centre for Cardiovascular Biology and Disease Research, Department of Microbiology, Anatomy, Physiology and Pharmacology, School of Agriculture, Biomedicine and Environment, La Trobe University, Melbourne, Australia; [2]Department of Physiology, Yong Loo Lin School Medicine, National University of Singapore, Singapore, Singapore; [3]School of Pharmacy, Sungkyunkwan University, Suwon, Republic of Korea; [4]Translational Biomedical Proteomics Laboratory, Institute of Molecular and Cell Biology, Agency for Science, Technology and Research, Singapore, Singapore; [5]National Heart Research Institute, National Heart Centre Singapore, Singapore, Singapore; [6]Institute for Molecular Medicine, University of Southern Denmark, Odense, Denmark; [7]National Institute of Pathology "Victor Babes", Bucharest, Romania; [8]NUS Graduate School for Integrative Sciences and Engineering, National University of Singapore, Singapore, Singapore; [9]Natural Products Research Center, Korea Institute of Science and Technology, Gangneung, Republic of Korea; [10]Translational Laboratories in Genetic Medicine, Agency for Science, Technology and Research, Singapore, Singapore; [11]Cardiovascular and Metabolic Disorders Program, Duke-National University of Singapore, Singapore, Singapore; [12]Department of Neurology, University Medical Center Hamburg-Eppendorf, Hamburg, Germany; [13]Department of Biochemistry, Yong Loo Lin School Medicine, National University of Singapore, Singapore, Singapore; [14]Department of Medicine, Yong Loo Lin School of Medicine, National University of Singapore, Singapore, Singapore; [15]Department of Neuroscience, Johns Hopkins University School of Medicine, Baltimore, United States; [16]Department of Anatomy, Yong Loo Lin School of Medicine, National University of Singapore, Singapore, Singapore

**Competing interest:** The authors declare that no competing interests exist.

**Abstract** Intermittent fasting (IF) has been shown to reduce cardiovascular risk factors in both animals and humans, and can protect the heart against ischemic injury in models of myocardial infarction. However, the underlying molecular mechanisms behind these effects remain unclear. To shed light on the molecular and cellular adaptations of the heart to IF, we conducted comprehensive system-wide analyses of the proteome, phosphoproteome, and transcriptome, followed by

functional analysis. Using advanced mass spectrometry, we profiled the proteome and phosphoproteome of heart tissues obtained from mice that were maintained on daily 12- or 16 hr fasting, every-other-day fasting, or ad libitum control feeding regimens for 6 months. We also performed RNA sequencing to evaluate whether the observed molecular responses to IF occur at the transcriptional or post-transcriptional levels. Our analyses revealed that IF significantly affected pathways that regulate cyclic GMP signaling, lipid and amino acid metabolism, cell adhesion, cell death, and inflammation. Furthermore, we found that the impact of IF on different metabolic processes varied depending on the length of the fasting regimen. Short IF regimens showed a higher correlation of pathway alteration, while longer IF regimens had an inverse correlation of metabolic processes such as fatty acid oxidation and immune processes. Additionally, functional echocardiographic analyses demonstrated that IF enhances stress-induced cardiac performance. Our systematic multi-omics study provides a molecular framework for understanding how IF impacts the heart's function and its vulnerability to injury and disease.

## eLife assessment

This study provides a **useful** catalog of the cardiac proteome and transcriptome in response to intermittent fasting. Although mechanistic integration is limited, the technical aspects have been executed in a **solid** way, and sufficient evidence is provided to support the main conclusions. Future work can build on this study to expand our understanding of the relationship between dietary perturbations and cardiac function.

## Introduction

In the developed world, the average calorie intake has steadily risen, as have associated age-related diseases (*Lahey and Khan, 2018*). Intermittent fasting (IF) eating patterns include frequent periods of 12–24 hr with little or no energy intake sufficient to trigger a metabolic switch from the utilization of liver-derived glucose to fatty acids and ketones. Such IF eating patterns increase health span and lifespan and delay or prevent age-associated diseases in animals (*de Cabo and Mattson, 2019*; *Mattson et al., 2017*). Rodents maintained on IF exhibit reduced resting heart rate and blood pressure, increased heart rate variability, and improved cardiovascular adaptations to stress (*Mager et al., 2006*; *Wan et al., 2003a*; *Wan et al., 2003b*; *Wan et al., 2014*). Age-related increases in oxidative stress, inflammation, and fibrosis in the heart are prevented by every-other-day (EOD) fasting (*Castello et al., 2011*). IF also improves health indicators and cardiovascular risk factors in human subjects (*Harvie et al., 2011*; *Harvie et al., 2013*; *Stein et al., 2012*; *Weiss and Fontana, 2011*). In a recent randomized controlled trial, alternate day fasting improved cardiovascular markers, reduced fat mass, fat-to-lean ratio, and increased levels of the ketone β-hydroxybutyrate (BHB) (*Stekovic et al., 2019*). IF protects the brain and heart against ischemic injury and enhances endurance in animal models (*Arumugam et al., 2010*; *Kim et al., 2018*; *Marosi et al., 2018*; *Mattson and Arumugam, 2018*; *Wan et al., 2010*). Animals maintained on EOD fasting prior to experimental myocardial infarction (MI) have a reduced myocardial infarct size compared to the ad libitum (AL) fed controls (*Ahmet et al., 2011*; *Ahmet et al., 2005*; *Godar et al., 2015*). IF also improves survival and recovery of heart function when initiated two weeks after MI (*Katare et al., 2009*). Recent brain and skeletal muscle studies suggest that IF engages signaling pathways that enhance cellular stress resistance, mitochondrial biogenesis, and autophagy (*Marosi et al., 2018*; *Mattson and Arumugam, 2018*). The ketone BHB may mediate some cellular adaptations to fasting, including inducing the expression of the inhibitory neurotransmitter γ-aminobutyric acid, trophic factors, and the cytoprotective mitochondrial deacetylase sirtuin 3 (*Cheng et al., 2016*; *Liu et al., 2019*; *Marosi et al., 2016*; *Newman and Verdin, 2017*; *Yudkoff et al., 2007*). However, it is unknown whether similar molecular and cellular mechanisms underlie the beneficial effects of IF on the cardiovascular system.

System-wide interrogation using large-scale omics technologies has proven powerful in capturing snapshots of molecular abundances in an unbiased manner. Previous studies have extensively focused on understanding the metabolic rewiring associated with different modes of fasting and feeding responses in the liver, the primary metabolic and secretory organ, and adipose tissues that functions as a fat sensor (*Ng et al., 2019*). Though transcriptomics remained the go-to technology for many of

the earlier works in this arena, recent technological advancements have positioned mass spectrometry (MS)-based proteomics as a rational choice to simultaneously track the dynamic behavior of thousands of proteins, the prime orchestrators of cellular processes, in a global manner. Accordingly, proteotypes underlying various metabolic diseases, dietary milieu, and fasting regimens including time-restricted feeding or IF have been probed to get holistic molecular portraits of altered metabolic states (*Shaik et al., 2016*; *Hatchwell et al., 2020*). Protein post-translational modifications, especially phosphorylation, underlie several energy sensor pathways, glucose homeostasis mechanisms, and insulin secretion process, and hence deconvolution of phosphorylation networks is key to understanding signaling alterations (*Sacco et al., 2019*; *Sacco et al., 2016*). While the importance of IF in improving cardiovascular health has been extensively documented, a global investigation documenting the molecular players driven by the metabolic switch would be highly desirable.

In order to understand molecular and cellular remodeling of the heart during IF, we employed advanced mass spectrometry for system-wide profiling of the proteome and phosphoproteome of heart tissues obtained from mice with daily fasting for 12 hr (IF12), 16 hr (IF16) or EOD as well as AL control feeding. We also performed transcriptome analyses using RNA sequencing to evaluate whether the observed molecular responses to IF occur at the transcriptional or post-transcriptional levels. Our results indicated that IF profoundly modifies pathways involved in metabolism, cell signaling, and epigenetic reprogramming in the heart. Of note, we observed that protein network remodeling during IF occurred in an IF duration time-dependent manner, with IF16 and EOD having the most significant effects. In addition, functional studies indicated that IF improves cardiac function compared to AL feeding. Our findings provide novel insight into the genetic and proteomic changes by which IF improves cardiac health and provide a resource for investigators in the fields of metabolism and cardiovascular disease.

## Materials and methods
### Animals and IF procedures

All in vivo experimental procedures were approved by the National University of Singapore (Ethics approval number: R15-1568) and La Trobe University (Ethics approval number: AEC21012) Animal Care and Use Committees and performed according to the guidelines set forth by the National Advisory Committee for Laboratory Animal Research (NACLAR), Singapore, and Australian Code for the Care and Use of Animals for Scientific Purposes (8th edition) and confirmed NIH Guide for the Care and Use of Laboratory Animals. All efforts were made to minimize suffering and the number of animals used. All sections of the manuscript were performed in accordance with ARRIVE guidelines. Animals were anaesthetized in 1–2% isoflurane, and anesthesia was confirmed by abolished pain reflexes. Animals were euthanized by cervical dislocation, and the chest was then opened to extract the heart. C57BL/6 male mice were purchased at 2 months of age (InVivos, Singapore; ARC, Australia) and housed in the animal facility at the National University of Singapore and La Trobe University. The animals were exposed to light from 07:00-19:00 (12 hr light: 12 hr dark cycle) and were fed with pellets for rodents which consisted of 58%, 24%, and 18% of calories originating from carbohydrate, protein, and fat, respectively (Teklad Global 18% Protein rodent diet #2918; Envigo, Madison, WI, USA). Water was provided ad libitum to all dietary groups. At the age of 3 months, the mice were randomly assigned to the four different dietary intervention groups: IF12 (N=20), IF16 (N=60), EOD (Every Other Day fasting) (N=20), and AL group as a control (N=60). The mice in the IF12, IF16, and EOD groups fasted for 6 months daily 12 hr (19:00-07:00), 16 hr (15:00-07:00), or 24 hr (a whole day fasting followed by an entire day feeding). The AL group was supplied with food pellets ad libitum. The body weight was regularly measured. Blood glucose and ketone levels using FreeStyle Optium Neo system with FreeStyle Optium blood glucose and ketone test strips (Abbott Laboratories, Berkshire, UK) were measured in the morning before IF animals received food (N=20 in each experimental group). After 6 months, animals were anesthetized and euthanized. EOD mice were euthanized on a food-deprivation day. All mice were euthanized under isoflurane anesthesia between 7 a.m. and noon. All mice were perfused with cold PBS. The chest cavity was rapidly opened, and the heart was removed and rinsed in two washes of ice-cold saline. Major blood vessels and connective tissue were removed, and the heart was blotted dry, and weighed. The heart samples were collected and kept at –80 °C until further use.

## Sample preparation for proteome analyses

Frozen tissue samples from all groups (Number of animals in AL = 4, IF12=5, IF16=5, EOD = 4) were crushed in liquid nitrogen prior to lysis with 6 M urea (Sigma-Aldrich), 2 M thiourea (Invitrogen), and 20 mM HEPES. The lysate was collected after centrifugation at 20,000 $g$ for 20 min at 15 °C, and protein measurement was made using Pierce 660 nm Protein Assay Reagent (Thermo Scientific) according to the manufacturer's protocol. Fifty μg of protein was used to prepare for each TMT label. For the reference channel of 126-TMT label, a total of 50 μg protein pooled from the 4 ALs was used. The reduction was performed with a final concentration 5 mM dithiothreitol (Sigma-Aldrich) incubation for 30 min. This was followed by alkylation with a final concentration of 10 mM iodoacetamide (Sigma-Aldrich) for 30 min incubation in the dark. Digestion was carried out using lysyl endopeptidase digestion (Wako Chemicals GmbH) with enzyme: protein = 1:100 at 37 °C overnight. The urea in the lysis buffer was then diluted to a final concentration of 1 M with 50 mM ammonium bicarbonate followed by trypsin digestion (Promega) with enzyme: protein = 1:50 at 37 °C for 8 hr. Digested peptides were desalted using 1 ml Empore C18 cartridges (3 M) before isobaric labeling.

## Sample preparation for phosphoproteome analysis

A total of three animals from each group were used for phosphoproteome analysis. Immobilized metal affinity chromatography (IMAC) was employed for the selective enrichment of phosphorylated peptides. Briefly, Ni-NTA agarose conjugates (Qiagen) were rinsed thrice with MilliQ water and incubated with 100 mM EDTA pH 8.0 for 30 min at RT to strip the nickel. Residual nickel was rinsed thrice with MilliQ water, followed by incubation with 100 mM iron chloride solution for 30–45 min. The resin was then washed thrice with MilliQ water followed by 80% acetonitrile / 0.1% trifluoroacetic acid. The tryptic peptides were then enriched using the iron chloride treated IMAC beads. Tryptic peptides were reconstituted in 50% acetonitrile/0.1% trifluoroacetic acid, followed by 1:1 dilution with 99.9% acetonitrile/0.1% trifluoroacetic acid. The peptides were incubated with 10 μL of IMAC beads for 30 min with end-over-end rotation, and the beads were subsequently loaded onto self-packed C18 stage tips (*Rappsilber et al., 2007*) pretreated with 200 μl methanol, washed with 50% acetonitrile/0.1% formic acid and equilibrated with 1% formic acid. Following loading onto the stage tip, the IMAC beads were washed with 80% acetonitrile/0.1% trifluoroacetic acid and 1% formic acid. The phosphopeptides were eluted from the IMAC beads onto C18 membranes using 500 mM dibasic sodium phosphate (pH 7.0), followed by washing using 1% formic acid. The phosphopeptides were stored on the stage tip until they were ready to be analysed by liquid chromatography-mass spectrometry (LC-MS). The phosphopeptides were eluted from the C18 membranes with 50% acetonitrile/0.1% formic acid, dried using a speed vac, and reconstituted in 0.1% formic acid for LC-MS analysis.

## Isobaric labeling

All samples were run in a total of two sets of TMT10plex Isobaric Label Reagent Set (Thermo Scientific) and were assigned to each TMT set (9 samples to 9 plexes and an internal reference sample made up of a mixture of the ALs) by randomization. Isobaric labelling was performed according to the manufacturer's protocol. Briefly, the TMT label was reconstituted in anhydrous acetonitrile (Sigma-Aldrich) before mixing with respective peptides and incubated for 1 hr. The reaction was quenched using the final 5% hydroxylamine (Merck) in triethylammonium hydrogen carbonate buffer (Fluka) for 15 min. Labeled peptides were mixed before being vacuum-dried and fractionated using basic reverse phase chromatography (Shimadzu NexeraXR high performance liquid chromatography). Fractions were combined orthogonally into 15 fractions and dried using a vacuum centrifuge.

## Liquid chromatography and mass spectrometry

For proteome analysis, peptides were reconstituted in 0.1% formic acid for analysis on Thermo Easy nLC 1000 that was connected to Q Exactive Plus Quadrupole-Orbitrap Mass Spectrometer (Thermo Scientific). The trap column used was C18 Acclaim PepMap 100 of 5 μm, 100 A, 100 μm I.D. x 2 cm, and the analytical column was PepMap RSLC C18, 2 μm, 100 A, 75 μm I.D. x 50 cm. The LC solvent A comprised of 0.1 M formic acid in 2% acetonitrile, and LC solvent B comprised 0.1 M formic acid in 95% acetonitrile. The gradient was as follows: 5–50% solvent B in 180 min; 50–100% solvent B in 10 min; 100% solvent B for 10 min at the flow rate of 200 nL/min. The mass spectrometer was set in the data dependent acquisition mode. Full scan MS spectra (m/z 380–1600) were acquired with a

resolution of $R$=140,000 at an AGC target of 3e6 and a maximum injection time of 60ms. The top 20 most intense peptide ions in each MS scan were sequentially isolated to an ACG target value of 1e5 with a resolution of 35,000 and fragmented using a normalized collision energy of 30 at MS2 level with fixed first mass at 110 and an isolation window of 1.2 m/z. For phosphoproteome, reconstituted peptides were analysed using an EASY-nLC 1000 (Proxeon, Thermo Fisher Scientific) attached to a Q-Exactive (Thermo Fisher Scientific). Peptides were enriched using a C18 precolumn and separated on a 50 cm analytical column (EASY-Spray Columns, Thermo Fisher Scientific) at 50 °C using a 265 min gradient ranging from 0 to 40% acetonitrile/0.1% formic acid, followed by a 10 min gradient ranging from 40 to 80% acetonitrile/0.1% formic acid and maintained for 10 min at 80% acetonitrile/0.1% formic acid. Survey full scan MS spectra (m/z 310–2000) were collected with a resolution of $r$=70,000,, an AGC target of 3e6, and a maximum injection time of 10ms. Twenty of the most intense peptide ions in each survey scan with an intensity threshold of 10,000, underfill ratio of 1%, and a charge state ≥2 were sequentially isolated with a window of 2 Th to a target value of 50,000 with a maximum injection time of 50ms. These were fragmented in the high energy collision cell by dissociation, using a normalized collision energy of 25%. The MS/MS was acquired with a starting mass of m/z 100 and a resolution of 17,500 and dynamic exclusion of duration of 15 s.

## Total RNA extraction and quality control for transcriptome analyses

EZ-10 Total RNA Mini-Prep Isolation kit (Bio Basic, Canada) was used according to the kit's instructions to extract total RNA from the heart samples. To assess the quality of the extracted total RNA, agarose gel electrophoresis and Agilent Bioanalyzer 2100 system (Agilent, Santa Clara, CA, USA) were used. All RNA samples (N=5 in each experimental group) had high quality with integrity numbers >6.9 for total RNA.

## cDNA library preparation and RNA sequencing

Poly-T oligo-attached magnetic beads were used to purify the mRNA from total RNA. Then, a fragmentation buffer was added to the mRNA. First-strand cDNA was subsequently synthesized using random hexamer primer and M-MuLV reverse transcriptase (RNase H-; New England BioLabs, Ipswich, MA, USA). Next, second-strand cRNA synthesis was performed using DNA polymerase I and RNase H. AMPure XP beads (Beckman Colter Life Sciences, Indianapolis, IN, USA) were used to purify the double-stranded cDNA. By using exonuclease/polymerase activities, the remaining overhangs of the purified double-stranded cDNA were transformed into blunt ends. The 3' ends of the DNA fragments were adenylated, followed by ligation of the NEBNext adaptor with a hairpin loop structure to prepare for hybridization. The library fragments were purified with the AMPure XP system to select cDNA with the base pair length of 150–200. Lastly, the library was amplified by polymerase chain reaction (PCR), and the PCR products were purified using AMPure XP beads. The HiSeq 2500 platform (Illumina, San Diego, CA, USA) was used to conduct the high-throughput sequencing.

## Transcriptome data mapping and differential expression analyses

The RNA sequencing results from the HiSeq system were output as quality and color space fasta files. The files were mapped to the Ensembl-released mouse genome sequence and annotation (*Mus musculus*; GRCm38/mm10). Bowtie v.2.0.6 Indexes were used to build the reference genome, and TopHat v.2.0.9 with a mismatch parameter limited to 2 was used to align the paired-end clean reads to the reference genome. A total of 48.5 million clean reads were generated per sample, and 43.7 million reads (90.1% of total clean reads) per sample were mapped to the reference genome. HTSeq v.0.6.1 was used for the quantification of gene expression levels to count the read numbers mapped of each gene. Then FPKMs of each gene were calculated based on the length of the gene and reads count mapped to the same gene. DESeq R Package v.1.10.1 was used to perform differential expression analysis. To control the false discovery rate, the p-values resulting from the analysis were adjusted to q-values using Benjamini and Hochberg's approach. Genes with q-value <0.05 found by DESeq were assigned as differentially expressed.

## Heat map generation and functional enrichment analyses

R and the R package heatmap3 were used along with the log2 fold change output from EdgeR v.3.2.4 to create heatmaps of DEGs. To estimate the possible biologic functional changes derived from the

gene expression changes, gene ontology (GO) and Kyoto Encyclopedia of Genes and Genomes (KEGG) pathway enrichment analyses were conducted. GO enrichment analysis was implemented by the GOseq R package, in which gene length bias was corrected. For KEGG pathway enrichment analysis, we used KEGG Orthology Based Annotation System (KOBAS) software to test the statistical enrichment. GO terms and KEGG pathways with a q-value less than 0.05 were considered significantly enriched by DEGs.

## Proteome and phosphoproteome data processing

For 10plex-TMT labeled proteome analysis, raw MS data were processed using Proteome Discoverer 2.2 (Thermo Scientific). Database search was performed using the integrated Sequest HT search engine against the Uniprot mouse FASTA database (release January 2018) for tryptic peptides with a maximum of two missed cleavage sites, MS and MS/MS mass tolerance of 10 ppm and 0.02 Da, respectively. Searches included cysteine carbamidomethylation and TMT-modifications at peptide N-termini and lysine residues as fixed modifications and protein N-terminal acetylation and methionine oxidation as dynamic modifications. Peptide and protein identifications were performed at a false discovery rate (FDR)<0.01. For phosphoproteome analysis, raw MS data were processed using MaxQuant version 1.6.0.1 (*Tyanova et al., 2016a*). The Andromeda search engine was employed to search against the UniProt mouse FASTA database. For identification of proteins, peptides, and modifications, the FDR was set at 0.01, allowing a maximum of two missed cleavages as well as an initial mass tolerance of 4.5 ppm for precursor ions and 20 ppm for fragment ions. The search was performed with phosphorylation of sites serine, threonine, and tyrosine (STY), oxidation of methionine, and acetylation of protein N-term as variable modification, cysteine carbamidomethylation as fixed modification, and trypsin as cleaving enzyme. Label-free quantitation (LFQ) was performed using the MaxLFQ algorithm as implemented in MaxQuant with minimum ratio count set to 1 and match between run feature enabled to transfer peptide identifications across MS runs (match time window = 2, alignment time window = 20) (*Cox et al., 2014*). A minimum Andromeda score of 40 was used for phosphopeptide identification.

## Proteome and phosphoproteome data analyses

For TMT-based proteome analysis, the common internal reference, pooled AL sample, at the 126 TMT-reporter channel of each TMT set was used for normalization across the two sets. The pooled ad libitum was included as an internal reference for the TMT sets as our focus was on inferring proteome abundance changes and associated biological pathways across the different IF regimens in comparison with the control ad libitum group. Protein abundances across all the channels were first normalized for equal total protein intensities before obtaining log2 ratios of each protein against the normalized intensity of the respective reference channel. After combining the two TMT sets, only those proteins that were quantified in at least three biological replicates within each group were retained for subsequent analysis. For differential expression analysis of proteins regulated with IF, the replicates from the two TMT sets were grouped, and one-way analysis of variance (ANOVA) was performed with followed by Dunnett's post hoc tests at adjusted p-value <0.01 to test for pairwise significance testing against AL control group. Protein annotations for gene ontology (GO) terms (biological process, cellular component, and molecular function) and KEGG pathways were obtained, and functional enrichment analysis was performed using Fisher's exact test (FDR ≤0.05) for the modulated clusters. In addition, 1-D annotation enrichment analysis as implemented in Perseus was performed for testing the preference of protein expression values to be higher or lower in comparison to the global distribution for specific functional groups (*Tyanova et al., 2016b*). The FDR was controlled at a threshold of 0.05, and for each functional group, a score was assigned, with scores those higher than zero and close to 1 confirming to positive enrichment, and those lesser than zero and close to –1 representing negative enrichment (*Subramanian et al., 2005*). To construct functional networks over the different IF regimens, gene-set enrichment analysis (GSEA) was performed on the proteome data using all curated mouse GO biological processes and pathways. Pairwise comparisons of the different groups were performed, and the statistical significance of the enrichment score was assessed by a permutation test. Based on the similarity between enriched gene sets (*P*-value <0.01), the EnrichmentMap plugin, as implemented in Cytoscape, was used to visualize all overlapping gene set clusters (*Merico et al., 2010*). All heat maps corresponding to functional categories were visualized using an internal

log-transformed reference normalized abundance value for each protein (*Plubell et al., 2017*). Transcription factor enrichment analysis for significant proteins in each IF group was performed using the Chip Enrichment Analysis (ChEA) implemented in the Enrichr web tool (*Kuleshov et al., 2016*; *Lachmann et al., 2010*).

For phosphoproteome analysis, the label-free intensities of all identified phosphopeptides were log-transformed, and only those phosphopeptides quantified in at least two biological replicates of each group were retained for all subsequent analyses. Using the Perseus software environment, missing values were imputed to represent intensities of low abundant proteins based on random numbers drawn from a normal distribution in each sample (*Tyanova et al., 2016b*). Phosphorylation sites differentially regulated between the groups were identified using one-way ANOVA followed by Dunnett's post hoc test at an adjusted p-value ≤0.05. Biological processes and KEGG pathways for differentially altered phosphoproteins were derived from DAVID bioinformatics resources (*Huang et al., 2009*). For upstream kinase enrichment analysis, kinase substrate motifs were extracted from the human protein reference database (HPRD), and enrichment analysis was performed using Fisher's exact test for altered phosphosite clusters across the groups.

## Comparison of proteome and transcriptome data

For the evaluation of the proteome against the transcriptome, only those transcripts with corresponding protein quantification were used. Correlation analysis was performed using FPKM as a proxy for mRNA abundance and normalized protein abundance derived from TMT experiments for protein quantification. Data were log-transformed prior to visualization. For comparison of transcriptome and proteome changes across different IF groups, log-transformed fold changes relative to AL group were used as input to perform two dimensional (2-D) functional enrichment analyses as implemented in Perseus (*Tyanova et al., 2016b*). The respective annotation matrix of KEGG pathways and GO terms (biological process) of both the transcriptome and proteome were compared for significant bias to larger or smaller expression values in comparison to overall distribution. The FDR was controlled at a 0.05 threshold and position score was assigned to denote enrichment at higher (0<score < 1) and lower (–1<score < 0) abundances. A scatter plot was constructed based on the positional scores, and functional categories were highlighted.

## Protein-protein interaction network analyses

Functional connectivity between significantly altered proteins was explored using protein-protein interactions curated in the Pathway Commons database, which compiles pathway and interaction data from multiple resources (*Cerami et al., 2011*). In addition to direct physical interaction between proteins, associations that represent functional interactions such as those that regulate expression, control protein state change, and catalysis, or mediate complex formation were also considered. The IF-altered functional network encompassed interactions relevant to all metabolic, regulatory as well as signaling pathways. Duplicate interactions were removed to remove redundancy, and the network was populated with information on the regimen-specificity of altered proteins to obtain time-resolved functional networks. Cluster assessment was performed using Cluster ONE implemented in Cytoscape to identify densely connected regions within the network (p-value <0.1; *Nepusz et al., 2012*; *Shannon et al., 2003*). The functional biases of the identified clusters for ontology and pathways were derived from DAVID bioinformatics resources (*Huang et al., 2009*). Finally, the topological parameters of the network were assessed for node centrality and degree distribution.

## Perturbation association network analyses

A correlation matrix was obtained based on the quantitative proteome from all groups using Spearman's rank correlation. The correlation coefficients and p values were calculated for all pairs of proteins, and multiple testing was controlled using Benjamini-Hochberg correction. Proteins pairs with adjusted p-value ≤0.01 were filtered to retain only those highly confident protein-protein associations (absolute correlation coefficient ~0.82) pertaining to both positive and negative correlations. Unweighted correlation networks were assembled using the high confident associations, and following the 'interactome mapping by high-throughput quantitative proteome analysis' (IMAHP) method (*Lapek et al., 2017*), specific deviations of each sample from the correlations were evaluated using Mahalanobis distance followed by outlier analysis using Grubbs's test (p-value ≤0.1) to identify

perturbed associations in each sample. To identify consensus-perturbed proteins among different dietary groups, a hypergeometric test was performed against the background of all perturbations (p-value ≤0.05). Functional assessment of perturbed proteins in each group for processes and pathways were derived from DAVID bioinformatics resources (*Huang et al., 2009*). For the construction of mitochondrial perturbation network, we extracted those perturbed proteins annotated "mitochondria" across all groups, along with their regimen-specific co-regulated associations to be visualized as a network with functional annotations.

## Immunoblot analysis

Frozen heart tissue samples (N=6 in each experimental group) were homogenized using a tissue tearor (BioSpec, Model: 985370) with protein and phosphatase inhibitors and Tissue Protein Extraction Reagent (Thermo Scientific, 78510) as the lysis buffer. The heart tissue samples were then combined with 2 x Laemelli buffer (Bio-Rad Laboratories, Inc, Hercules, CA, USA). Protein samples were then separated on 7.5–20% v/v sodium dodecyl sulfate (SDS) gels. Proteins in the SDS-PAGE gels were transferred onto nitrocellulose membranes to probe for proteins. Next, the nitrocellulose membranes were incubated with the following primary antibodies: p-AMPKα (2535 S, Cell Signaling), AMPKα (2532, Cell Signaling), PKG-1 (3248 S, Cell Signaling), MYPT1 (2634 S, Cell Signaling), PPARγ (7273, Santa Cruz Biotechnology), p-GSK3α/β (9331 S, Cell Signaling), GSK3α/β (7291, Santa Cruz Biotechnology), AKT (9272 S, Cell Signaling), Insulin receptor β (3025 S, Cell Signaling), Phospho-IRS1 (Ser318) (5610, Cell Signaling), IRS-1 (2382 S, Cell Signaling), and GAPDH (2118 S, Cell Signaling; NB300-221, Novus Biologicals) overnight at 4 °C with agitation. After primary antibody incubation, membranes were washed three times with 1xTBST before incubating with horseradish peroxidase (HRP)-conjugated secondary antibodies (Goat Anti-Rabbit – Cell Signalling Technology, Danvers, MA, USA; Goat Anti-Mouse – Sigma-Aldrich, St. Louis, MO, USA) for 1 hr at 24 °C with agitation. The membranes were then washed three times (10 min each) with 1xTBST. The substrate for HRP, enhanced chemiluminescence (ECL) (Bio-Rad Laboratories, Inc, Hercules, CA, USA), was applied for 3 min before the membranes were imaged using the ChemiDocXRS +imaging system (Bio-Rad Laboratories, Inc, Hercules, CA, USA). Quantification of proteins was conducted using Image J software (Version 1.52 a; National Institute of Health, Bethesda, MD, USA), where protein densitometry was expressed relative to the densitometry of the corresponding GAPDH respectively.

## Echocardiographic assessment of cardiac function

At the end-point of the experiment, myocardial function and cardiac dynamics were investigated by ultra-high frequency ultrasound for small animals (Vevo 3100 Preclinical Imaging System, FujiFilm VisualSonics, Canada) using the MicroScan Transducer MX550D (32-55MHz, Centre Transmit 40 MHz, Axial Resolution 40 µm) as previously described (*Liehn et al., 2013*; *Tyrankiewicz et al., 2013*). The Biomedical Sciences Institute Singapore Institutional Animal Care Committee approved all experiments. Mice were anesthetized with 1–2% isoflurane via mask and placed in supine position on a warming pad with integrated electrodes for continuous ECG monitoring (N=10 in each experimental group). Echocardiographic assessments of cardiac function were done in the morning and the IF mice were not fed prior to the experiments. Body temperature was monitored via a rectal temperature probe. After the complete removal of the hairs (hair removal cream), warmed up ultrasound gel was spread on the chest to avoid the formation of air bubbles and to ensure optimal sound quality. If needed, isoflurane was adjusted during the acquisition to maintain a relatively constant heart rate of over 300 beats per min (bmp). Two-dimensional and M-Mode were acquired in long and short axis using ECG-triggered image acquisition to assess ejection fraction, fractional shortening, cardiac output, stroke volume, and ventricular dimensions. The average values obtained from the long and short axis acquisitions were considered for all parameters. Strain analysis was performed on the long axis to measure the global longitudinal strain (GLS), strain rate, and radial strain (*Lindsey et al., 2018*). Pulsed wave (PW) Doppler was used to measure the velocity of blood flow through the aortic, pulmonary, mitral, and tricuspidal valves. Septal mitral annular velocity was measured using tissue doppler echocardiography. To uncover a potential cardiac functional reserve, dobutamine-induced stress was performed after basal parameter acquisition. Dobutamine was administrated intraperitoneally in a single injection at 20 mg/kg. After the maximum increase in heart rate (generally within 4–5 min), two-dimensional and M-Mode acquisitions in long and short axes were repeated to assess ejection

fraction, fractional shortening, cardiac output, stroke volume, ventricular dimensions, and strain in stress conditions.

## Statistics

For physiological measurements such as fasting blood glucose, ketone levels and body weights, values are indicated as mean ± SEM (n=20 mice/group). Significance is based on one-way ANOVA with Tukey's *post-hoc* test. *P*-value <0.05 was considered statistically significant. The statistics for these comparisons were conducted using the GraphPad Prism software programme. For transcriptome data, differential expression analysis of the quantified transcripts was performed using DESeq R Package v.1.10.1. To control for false discovery rate (FDR), the resulting p values obtained by the Wald test were adjusted to q-values (multiple testing) using the Benjamini and Hochberg's method. Those transcripts with q-value <0.05 were considered to be significantly differentially expressed among the different categories. For these significant genes, functional enrichment based on GO terms and associated KEGG pathways were inferred based on adjusted p-values <0.05 as implemented in KOBAS. For proteome and phosphoproteome analyses, all peptide and protein identifications together with their modifications (phosphosites) were performed at FDR <0.01 as implemented in Proteome Discoverer 2.2 (for proteome) and MaxQuant version 1.6.0.1 (for phosphoproteome). For TMT-based proteome data, biological replicates from the two TMT sets were combined following normalization and subjected to one-way analysis of variance (ANOVA) with Dunnett's post hoc test (multiple comparisons to control) at adjusted p-value <0.01 for pairwise significance testing against AL control group to correct for multiple testing. The statistical analyses were implemented in R statistical environment. For functional enrichment analyses using proteome data, Fisher's exact test was implemented, and enriched categories were identified at FDR ≤0.05. 1-D and 2-D annotation enrichment for the proteome profiles to assess the distribution of specific functional groups among the different IF categories were performed using Perseus and controlled for FDR at an adjusted p-value <0.05. For extracting differentially altered phosphorylation sites between the different groups, one-way ANOVA followed by Dunnett's *post-hoc* (multiple comparisons to control) was carried out and significantly altered sites were identified at adjusted p-value ≤0.05 to correct for multiple testing (implemented in R statistical environment). For immunoblot and echocardiographic analysis, all statistical analyses were conducted using the GraphPad Prism software programme. Data are presented as mean ± standard deviation for the indicated number of experiments. The difference in means between two columns was evaluated using the two-tailed unpaired Student's *t*-test (for two columns) or one-way ANOVA with Tukey's post-hoc test (for multiple comparisons to control). A p-value of less than 0.05 was considered statistically significant.

## Results

Male C57BL/6 mice were fed a normal chow diet comprised on a caloric basis of 58%, 24%, and 18% carbohydrate, protein, and fat, respectively. Mice were randomly assigned to AL, daily IF12, IF16, or EOD schedules beginning at 3 months of age. The study design, including the timing of experimental interventions and blood and tissue collections, is summarized in *Figure 1—figure supplement 1*. To determine the extent to which IF affects energy metabolism, we measured the blood glucose and ketone levels, and body weight of all mice during the six-month dietary intervention period. Mice in the AL group exhibited a trend toward increased blood glucose levels at 3 and 6 months. In contrast, mice in the EOD group exhibited a large and highly significant decrease in blood glucose levels throughout the duration of the study (*Figure 1—figure supplement 1b*). Mice in the IF12 and IF16 groups exhibited glucose levels intermediate to the AL and EOD groups (*Figure 1—figure supplement 1b*). Ketone levels also displayed variability across different fasting regimens. On fasting days, mice in the EOD group exhibited blood ketone levels that were threefold greater than mice in the AL group. Ketone levels in mice in the IF16 and IF12 groups tended to be higher than those in the AL group but were substantially lower than in the EOD group. Notably, all IF and EOD animals exhibited significantly lower body weight than AL mice (*Figure 1—figure supplement 1c*). However, despite the differences in the fasting regimen pattern, we did not observe any difference in overall body weight between the IF and EOD groups (*Figure 1—figure supplement 1c*). Furthermore, no

significant difference in the overall energy intake or compositional intake is observed across mice subjected to IF groups compared to the AL group (*Figure 1—figure supplement 1d*).

## IF modifies the heart proteome

To obtain a comprehensive perspective of the mechanisms impacted by IF in heart, we carried out proteome profiling of heart tissues from at least four biological replicates comprising each dietary group (AL = 4, IF12=5, IF16=5, EOD = 4). The AL group served as a control to compare proteomic alterations across different IF regimens. Using tandem mass tag (TMT)-based high resolution liquid chromatography-mass spectrometry (LC-MS) analysis, we identified a total of 6972 proteins (*Supplementary file 1*). Among the 5578 proteins quantified across all the samples, 9% of the proteins were differentially expressed among the heart tissues from different dietary regimens compared to the control AL group (Dunnett adjusted p-value = 0.01, *Supplementary file 2*). Among these, only 60 proteins consistently overlapped with altered protein levels across all fasting regimens and were primarily involved in processes associated with glycogen catabolism, fatty acid oxidation, and glycolysis (*Figure 1—figure supplement 2a*). In comparison with IF12, the proteomes were modulated to a greater extent in the longer fasting durations, IF16 and EOD. The EOD group exhibited the most significant number of protein abundance changes, with 44% modulations exclusively occurring within this group, including apolipoproteins that are important determinants of a healthy heart (*Liehn et al., 2018*). Approximately one-third of the differentially expressed proteins in IF groups compared to AL were enzymes with catalytic activity involved in energy homeostasis pathways. In addition to the expected fatty acid metabolism and glycolytic pathways, we observed enzymes involved in hypoxia, oxidative phosphorylation, bile acid metabolism, mTOR signaling, and myogenesis to be modulated with IF (*Figure 1—figure supplement 2b*). We observed that most of the enzymes involved in glycolysis, including the rate-limiting enzyme phosphofructokinase (PFKFKB1/2) displayed decreased expression with IF except for hexokinase 1 (HK1), indicative of suppression of glycolysis (*Supplementary file 2*). Conversely, the expression of phosphoenolpyruvate carboxykinase (PCK2), which is involved in gluconeogenesis, was enhanced by IF.

Unsupervised hierarchical clustering of the altered heart proteome revealed that IF12, IF16, and EOD mice had different overall protein expression patterns and co-expressed protein clusters compared to mice in the AL group (*Figure 1a*). Unlike clusters in IF16 and IF12 groups, which exhibited modest differences compared to the AL group, a distinct class of proteins showed increased abundance only in the EOD group. These clusters of proteins with exclusive upregulation in the EOD (FDR <0.05) group belonged to processes such as mRNA processing, fatty acid metabolism, autophagy, and stress response mechanisms. Other processes such as cholesterol transport and phospholipid efflux showed a downward trend in the EOD group. Several other metabolic processes, including oxidative phosphorylation, amine catabolism, and fatty acid oxidation, were overall reduced in all IF groups compared to AL. We also observed distinct modulations of several kinases in a regimen-specific manner, with majority of the alterations occurring only with IF16 and EOD (*Figure 1—figure supplement 2c*). Several of the kinases involved in the regulation of key metabolic pathways, such as glycolysis, exhibited decreased abundance in comparison with the AL group (*Figure 1b*). With EOD, major determinants of insulin sensitivity such as AKT1 and mTOR were highly expressed, and several other stress-activated kinases MAPK9 and MAPK10, showed reduced abundance (*Figure 1b* and *Figure 1—figure supplement 2c*).

To understand the molecular events accompanying each IF regimen, we performed 1-D annotation enrichment (*Cox and Mann, 2012*) identifying processes enriched at higher and lower protein abundances (*Figure 1c* and *Supplementary file 3*). We observed upregulation of structural proteins involved processes such as focal adhesion and cell adhesion across all fasting regimens suggesting possible re-organization of the extracellular matrix interaction in heart tissues with IF. In addition, proteins regulating lipid kinase activity were observed in higher abundance in the hearts of mice from all three IF regimens compared to AL. We also observed specific signaling pathways to be selectively enriched in one IF regimen compared to the other IF regimens. For example, the IF12 group showed enrichment of proteins associated with cyclase activity, while the IF16 group displayed elevation of ERK signaling proteins. Signaling pathways involving the second messenger cAMP, however, were activated only with longer periods of fasting. This supports the notion that different IF regimens differentially modulate protein expression changes in the heart and impact signaling rewiring (*Figure 1c*

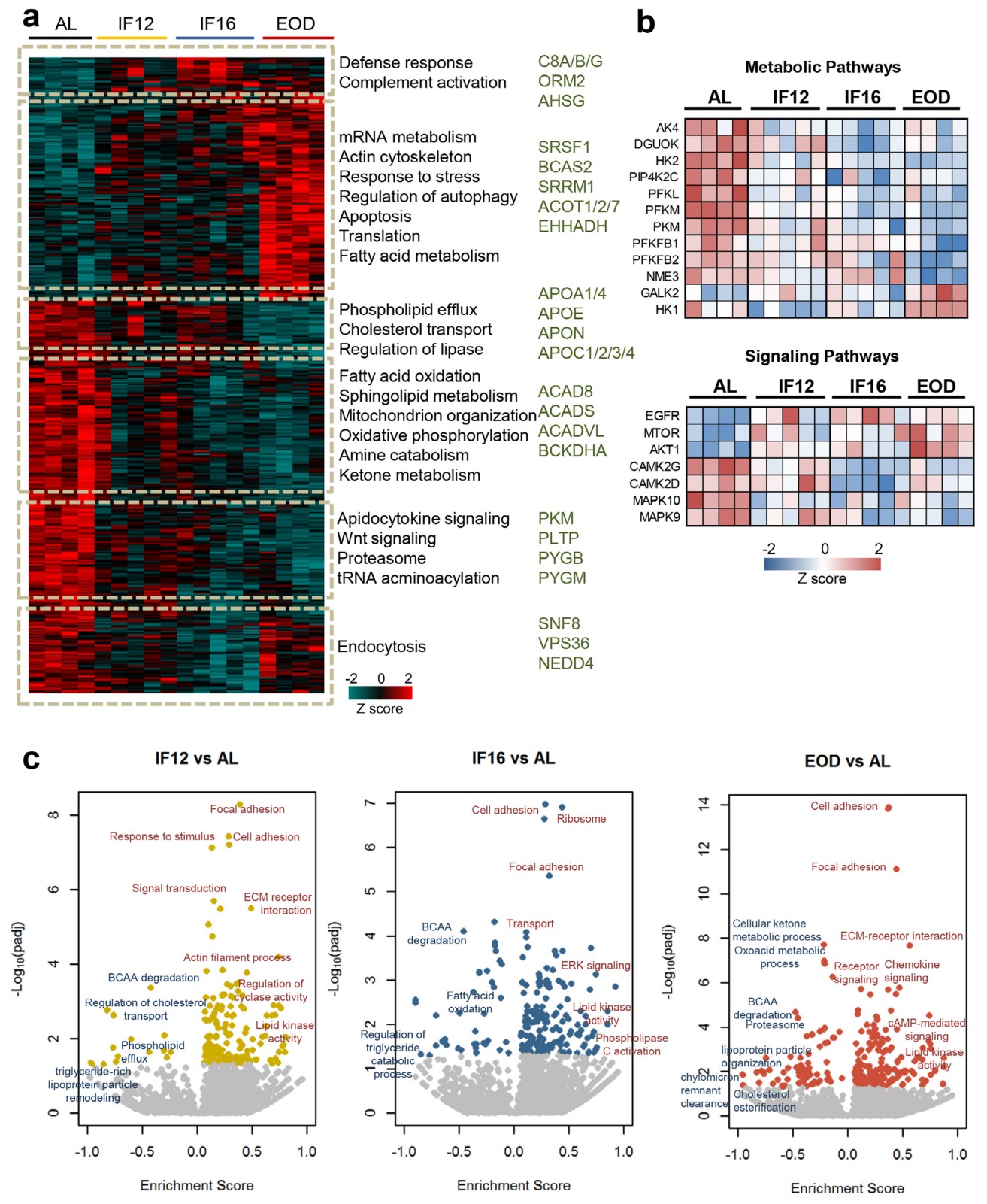

**Figure 1.** Impact of intermittent fasting on the heart proteome. (**a**) Hierarchical clustering of proteins significantly differentially expressed in each IF group and AL group (one-way ANOVA with Dunnett's test). The color scale denotes z-score normalized protein abundances. Red denotes higher protein abundance, and blue lower protein abundance. Gene ontology-based biological processes and pathways significantly enriched in representative protein clusters across the different groups. Selected enzymes specifically modulated within each cluster are highlighted. (**b**) IF-modulated kinases

*Figure 1 continued on next page*

*Figure 1 continued*

involved in metabolic pathways and signaling pathways. (**c**) Enrichment analysis of modulated processes and pathways in the different IF groups in comparison to the AL group (1-D annotation enrichment using Wilcoxon-Mann-Whitney test). Each dot represents a functional process or pathway. A positive enrichment score denotes higher protein abundance, and a negative enrichment score denotes lower protein abundance. n=4–5 mice in each experimental group.

The online version of this article includes the following figure supplement(s) for figure 1:

**Figure supplement 1.** Experimental design and details of the study cohort.

**Figure supplement 2.** IF-responsive proteins distribution across different fasting regimens.

and *Supplementary file 3*). Apart from components of lipid metabolism and transport that were reduced across all IF groups, our data also showed that branched-chain amino acid (BCAA) metabolism, which is known to regulate several anabolic and catabolic cellular cascades (*Gannon et al., 2018*), was significantly lower in all IF groups compared to the AL group (*Figure 1c*).

## Regimen-specific modulation of heart functional processes

The distinct and overlapping proteomic profiles for the three IF regimens suggested a progressive molecular and cellular remodeling of the heart in response to increasing fasting durations. While the overall metabolic responses were modulated across all regimens in comparison with AL, we further probed into regimen-specific differences to understand selective processes that were rewired with an increase in fasting duration (*Figure 2a–c*). Compared to AL, IF12 enhanced the processes that maintain the structural integrity of the heart, such as adhesion, extracellular structure, and cytoskeletal organization (*Figure 2a*). We also observed increased expression of proteins involved in the immune and stress response, and endocytosis and exocytosis events in IF12. Along with significant changes to the circulatory process, which displayed increased protein abundances, protein networks associated with heart rate regulation were also altered by IF12. As expected, proteins involved in metabolic processes such as sterol metabolism and transport, glucose utilization, and protein synthesis were downregulated (*Figure 2a*). While the protein trafficking network was enhanced in IF16, other immune functional networks (NF-κB and Wnt signaling, innate immune response, and T cell receptor signaling) and amino acid biosynthetic processes were downregulated in IF16 in comparison with IF12 (*Figure 2b*). Prolonged activation of NF-κB in immune cells occurs in several major cardiovascular diseases, including cardiac hypertrophy and heart failure, and aberrant Wnt signaling contributes to cardiac hypertrophy (*Foulquier et al., 2018*; *Gordon et al., 2011*). Changes in ion homeostasis and cardiac morphogenesis networks were also evident with an increase in fasting duration from IF12 to IF16. EOD introduced a wide spectrum of changes in the heart proteome. In addition to metabolic changes commonly, EOD led to epigenetic changes compared to IF16 (*Figure 2c*). Proteins associated with chromatin remodeling and organization, and those involved in histone and other protein modifications showed increased abundance. In parallel, proteins that regulate transcription initiation and termination, transcript splicing, and nuclear import were also upregulated in the hearts of EOD mice. Also, an increase in the mitotic and nuclear division regulatory networks was observed. This starkly contrasted to reductions in cell cycle-related pathways observed specifically in IF12 or IF16 groups. Additionally, DNA repair and organelle assembly protein networks were upregulated by EOD. Along with reduced immune response and ion and sterol transport, EOD exhibited a reduced abundance of components involved in mitochondrial electron transport chain (ETC), pyruvate metabolism, and ATP generation compared to IF16 mice (*Figure 2c*). EOD fasting also downregulated blood coagulation which may be crucial to reducing risk of pathological clot formation.

## Post-transcriptional buffering of IF-responsive proteins

While we established that IF impacts the heart proteome and relays a regimen-specific effect on the expression patterns of proteins, we next investigated the transcriptome to probe IF-induced effects. For this, we performed RNA-seq based transcriptome profiling for heart tissues obtained from mice in different IF regimens (*Figure 2—figure supplement 1* and *Supplementary file 4*). We obtained corresponding transcript readouts using RNA sequencing for approximately 93% of the proteins that were quantified across the different regimens. Comparison of correlation across transcripts and proteins abundance across the different groups revealed only a modest correlation as

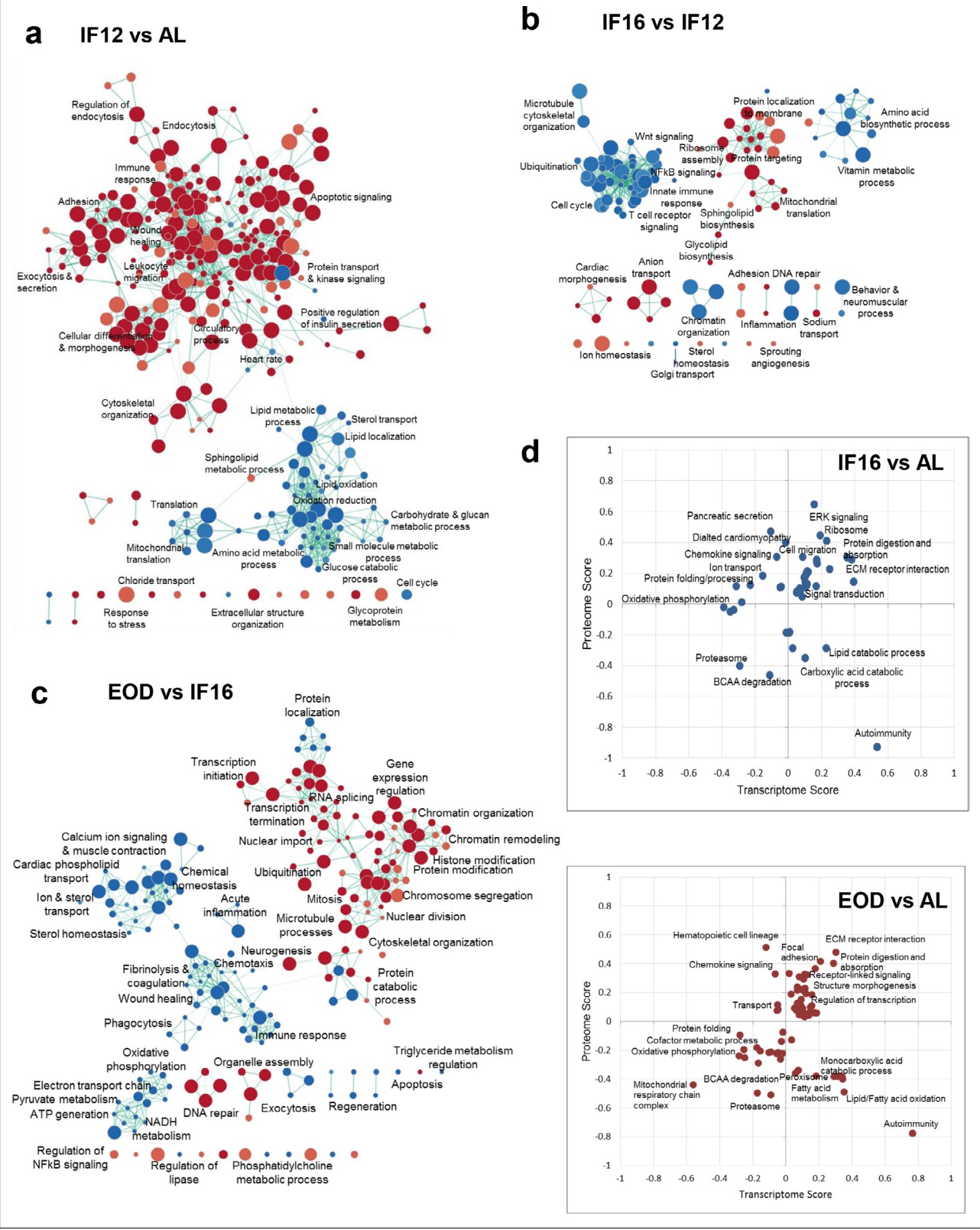

**Figure 2.** Differential effects of IF regimen functional processes. (**a**) Functional transition network that shows functional remodeling of heart proteome in response to IF12 (**a**), IF16 (**b**), and EOD (**c**) GSEA for biological processes and pathways performed for each IF regimen comparison is shown as an enrichment map. Functions significantly enriched (p<0.01, nominal p-value from empirical phenotype-based permutation test in GSEA) among the upregulated and downregulated proteins for each comparison are indicated by red and blue nodes, respectively. The size of the nodes indicates the

*Figure 2 continued on next page*

*Figure 2 continued*

number of proteins within the node and edges link nodes that share common proteins. (**d**) Functional processes and pathways altered by changes in transcriptome and proteome abundances through different IF regimens (Benjamini-Hochberg FDR <0.05, 2-D annotation enrichment using MANOVA test). Each dot on the plot represents a modulation in pathway or process. The scores indicate abundance changes of transcriptome or proteome levels over AL. Upregulation and downregulation of components are denoted by positive and negative scores, respectively. n=4–5 mice in each experimental group.

The online version of this article includes the following figure supplement(s) for figure 2:

**Figure supplement 1.** Transcriptomics-based analysis.

**Figure supplement 2.** Comparison of IF-induced transcriptome and proteome rewiring.

**Figure supplement 3.** Analysis of the core clock components genes.

expected (Pearson correlation coefficient ~0.47; *Figure 2—figure supplement 2a*), ruling out any regimen-specific differences. We next performed 2-D enrichment analyses to reveal correlated and non-correlated functional processes at the proteome and transcriptome levels (*Supplementary file 5*). This revealed that structural processes such as ECM-receptor interaction and metabolic pathways involving BCAA degradation showed consistent changes in proteome and transcriptome with IF (*Figure 2d* and *Figure 2—figure supplement 2b*). Nevertheless, with increased fasting hours, mainly in the EOD group, we observed a trend shift in the direction of proteome and transcriptome abundances (*Figure 2d*). Metabolic processes associated with fatty acid oxidation and monocarboxylic acid catabolic processes and immune processes related to autoimmunity and chemokine signaling showed the most noticeable drifts. On investigating the fatty acid oxidation cluster, we noticed that many of the key enzymes involved in the beta-oxidation of unsaturated (ECI1/2) and long-chain (or very long-chain) fatty acids (HADHA/B and ACADVL) showed differential transcript and protein abundances suggesting plausible buffering of protein levels beyond transcriptional regulation with IF. In comparing the enzyme abundances (*Figure 2—figure supplement 2c*), we found that some enzymes showed conserved correlation across different IF groups implying concerted rewiring of both transcriptome and proteome (*Figure 2—figure supplement 2c*, *Supplementary file 2*, and *Supplementary file 4*). Phospholipid phosphatase 1 (PLPP1) which is involved in the synthesis of glycerolipids and lipid uptake showed a dramatic decrease in protein abundance in the EOD group despite a modest increase in the corresponding transcript abundance. N-acetyltransferase NAT10 involved in histone acetylation and chromatin organization, ribonuclease RNASE4 participating in mRNA cleavage, and spliceosome-associated protein CWC27 involved in mRNA splicing all showed an exclusive increase only in the protein level with no observed changes in the mRNA level. Changes in these proteins were more obvious in the EOD group, and this supports extensive gene regulatory rewiring with extended fasting duration. Such discordant expression patterns at the mRNA and protein level could arise from post-transcriptional buffering that could potentially impact enzymes and kinases abundance and functions via post-translational modifications affecting protein stability or allosteric regulations that can alter activity.

With changes in metabolism closely linked to and regulated by circadian clocks, we assessed if time-restricted feeding regimens of IF12 and IF16 mice during the day (light phase) displayed any altered circadian patterns compared to AL or EOD mice by analyzing expression changes of core clock components at both the transcript and the protein levels (*Figure 2—figure supplement 3*). This analysis showed a differential expression profile of most clock gene transcripts compared with the AL control group irrespective of the fasting regimen pattern (*Supplementary file 4*). Notably, both the IF12 and IF16 groups were similar to the EOD fasting group despite the time-restricted feeding during daytime. Except for the transcripts of *CRY1* and *NR1D1* that showed differential expression between IF12 and EOD groups, all other clock gene transcripts showed consistent expression patterns across the three IF groups. In the corresponding proteome data, we only reliably quantified two clock components, CLOCK, and CRY1, neither of which showed significant differential protein expression among the fasting regimens.

## Time-resolved protein network remodeling during IF

To elucidate how functional protein networks are remodeled in response to IF, we curated protein-protein interactions (PPI) of all IF-responsive proteins and explored their functional crosstalk (*Figure 3*,

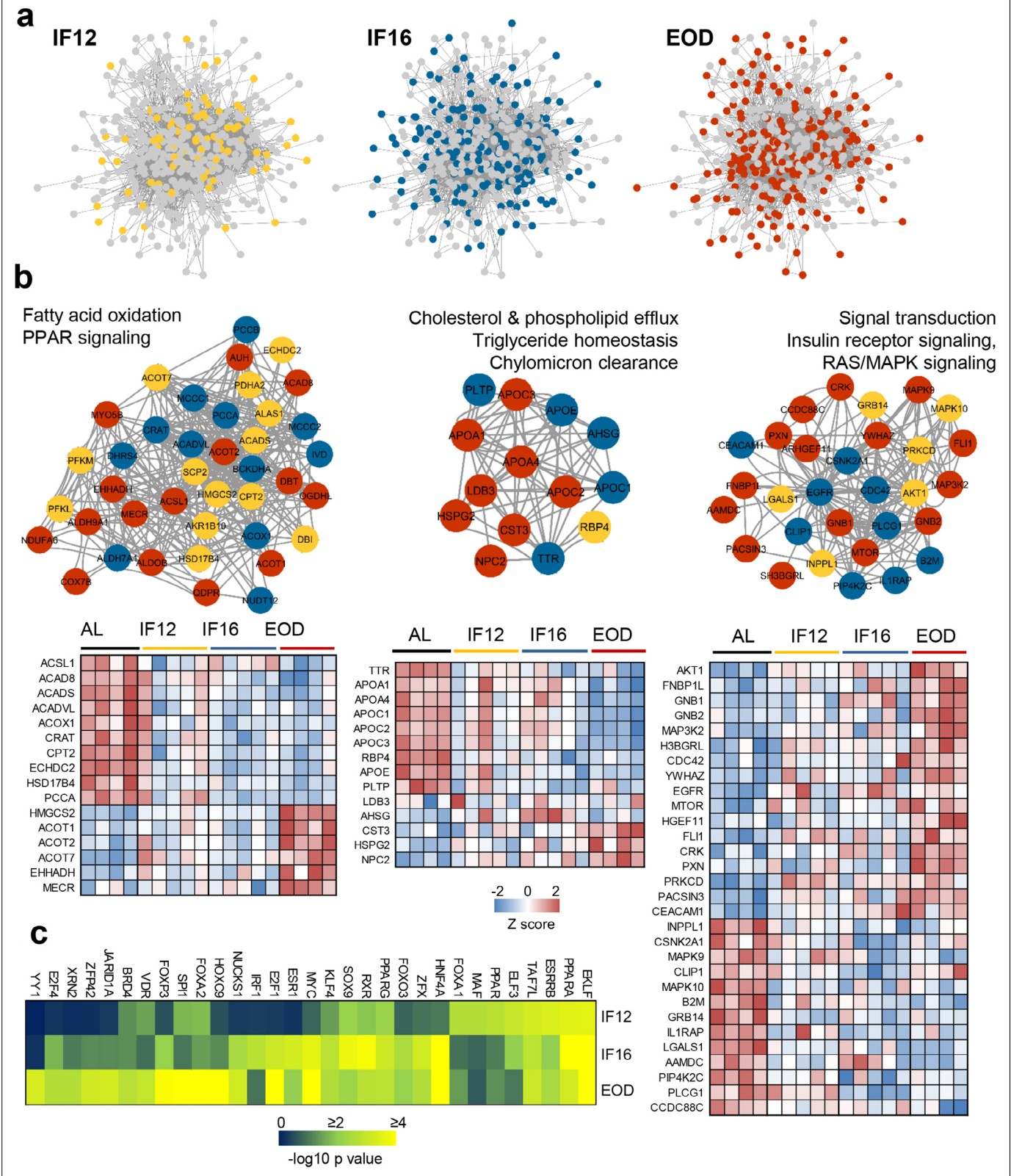

**Figure 3.** Network mapping of IF-responsive proteins. (**a**) Proteins with significantly altered abundance in response to IF are overlaid on consensus protein-protein interaction network constructed using all IF-responsive proteins. The proteins that sequentially show differential modulation are indicated with different colors as yellow (IF12), blue (IF16), and red (EOD). (**b**) The top densely connected functional protein clusters are shown (one-sided Mann-Whitney U test as implemented in Cluster ONE). The color of each node indicates the IF regimen for which the protein was differentially

*Figure 3 continued on next page*

*Figure 3 continued*

expressed. Yellow, blue, and red indicate IF12, IF16, and EOD regimens, respectively. The heat maps show proteome changes for the proteins in the respective clusters. Red denotes increased expression, and blue denotes decreased expression. (c) Transcription factors significantly enriched across the different IF groups. n=4–5 mice in each experimental group.

The online version of this article includes the following figure supplement(s) for figure 3:

**Figure supplement 1.** Network mapping of IF-responsive proteins across different regimens.

**Figure supplement 2.** Functional clusters enriched in IF-modulated network.

**Figure supplement 3.** Association perturbations across different IF regimens.

**Figure supplement 4.** A mitochondrial association network.

*Figure 3—figure supplement 1* and *Supplementary file 6*). Since many of the altered proteins were enzymes, we also considered functional interactions such as those that regulate gene expression, mediate enzymatic cascades or regulate protein phosphorylation in addition to physical interactions. To this end, we used the Pathway Commons database, a compendium of pathway-centric biologically relevant interactions (*Cerami et al., 2011*), and mapped interactions for 441 (81%) of the IF-responsive proteins. To infer duration-specific effects, we specifically marked those proteins that were first altered with shorter fasting duration and those that appeared with longer fasting hours, such as IF16 or EOD (*Figure 3a*). Changes in around 79 proteins appeared early within the IF12 regimen and some of these alterations were conserved until longer fasting durations at both IF16 and EOD (*Supplementary file 6*). Highly populated network clusters exclusively emerged with EOD that contributed to 189 proteins in the overall IF-modulated network (*Figure 3a*), suggesting enhanced network rewiring with longer fasting duration.

We next performed a cluster assessment using the Cluster ONE graph-clustering algorithm (*Nepusz et al., 2012*) to identify densely connected network regions that may be functionally relevant. This highlighted several tightly connected clusters that included proteins responsive to different IF regimens and showed functional preference for specific metabolic processes (*Figure 3b*, *Figure 3—figure supplement 2*, and *Supplementary file 6*). The cluster that constituted proteins associated with fatty acid beta-oxidation and PPAR was the most highly enriched cluster exemplifying the impact of IF on lipid oxidation. It was evident that many of the proteins associated with this cluster tend to be downregulated during IF, particularly in EOD, except for a few proteins such as ACOT1, ACOT2, and EHHADH that showed high abundances exclusively with EOD (*Figure 3b*). Indeed, cardioprotective roles have been attributed to the fatty acyl-CoA thioesterase ACOT1 via its activation of PPARα/PGC1α signaling (*Yang et al., 2012*). Another lipid metabolism-associated cluster impacted by IF12, IF16, and more by EOD, included proteins involved in cholesterol efflux, triglyceride homeostasis, and chylomicron clearance (*Figure 3b*). The most prominent effects of IF on lipid metabolism were on the apolipoproteins APOA1/4, APOE, and APOC1/2/3, all of which declined in the EOD group. An increase in the level of NPC2, an intracellular transporter mediating the export of cholesterol from lysosomes (*Pfeffer, 2019*), was observed in the EOD group. PLTP, the increased activity of which is known to be associated with atherosclerosis, systolic dysfunction, obesity, and diabetes (*Chowaniec and Skoczyńska, 2018*), was observed to be suppressed explicitly in all three IF groups.

Apart from these lipid-linked clusters, we also observed other clusters differentially modulated in an IF group-dependent manner. Signal transduction primarily via the insulin receptor and MAPK pathways emerged as one of the prominent clusters primarily comprising proteins modulated with longer fasting duration (*Figure 3b*). B2M, a risk marker for cardiovascular disease (*Shi et al., 2021*), was found within this cluster. The network comprising RNA processing and splicing proteins mostly constituted proteins changing with EOD (*Figure 3—figure supplement 2* and *Supplementary file 6*). The cluster with proteins involved in ubiquitination and proteolytic processes exhibited mostly included proteins altered with IF16. Upregulated by IF16 within this cluster was LNPEP, a zinc-dependent aminopeptidase that cleaves several peptide hormones and also catalyzes the conversion of angiotensinogen to cardioprotective angiotensin IV (AT4) (*Stragier et al., 2008*; *Yang et al., 2011*). From the network, we infer that other metabolic processes such as pyruvate and amino acid metabolism and glycolysis show gradual modulations in protein abundance with increasing fasting duration (IF12<IF16<EOD) (*Figure 3—figure supplement 2*). Further, based on their centrality, we inferred kinases including

AKT1, CDC42, PKM, EGFR, and CSNK2A1 as IF-responsive hubs that effectively relay signaling modulations during IF-induced network rewiring.

To determine how the IF-induced network rewiring impacts downstream transcriptional programs, we assessed transcription factors enriched at each IF regimen using ChIP enrichment analysis (ChEA) database (*Figure 3c*). Several key transcription factors with metabolic relevance including PPARα/γ, FOXA1/2, NUCKS1, HNF4A among others were found to be enriched. While some such as ELKF, TAF7L, and PPARα were modulated irrespective of the IF regimen, others, including HOXC9, YY1, and JARID1A showed exclusive enrichment in the EOD group.

## Modified protein co-regulation in heart during IF

Dynamic interactions among proteins coordinate cellular homeostasis and adaptive responses to environmental challenges. To decipher the effects of IF on such protein associations, we investigated the protein co-regulation network of heart proteome for IF regimen-specific perturbations using the 'interactome mapping by high-throughput quantitative proteome analysis' (IMAHP) method (*Lapek et al., 2017*; *Kosok et al., 2020*). The correlation network comprised 44,966 associating protein pairs, of which a majority (34,629 pairs) involved positive co-regulations between the associated proteins (*Figure 3—figure supplement 3* and *Supplementary file 7a-b*). To identify protein associations perturbed in an IF group-dependent manner, we carried out an outlier analysis to specifically derive associations that displayed significant deviation from the overall correlation across all co-regulated proteins. This revealed higher deviations by ~40% across all IF groups than the AL group, suggestive of dynamic modulation of protein associations with IF. We identified consensus proteins that were consistently altered in protein associations across the different IF regimens (*P*-value <0.05) (*Figure 3—figure supplement 3* and *Supplementary file 7c*) and accordingly identified the greatest number of affected proteins in the IF16 and EOD groups, with the EOD group showing exclusive changes in 118 proteins and the IF16 group with 69 altered proteins. The IF12 group displayed only a small number of perturbed associations similar to the AL group. Also, we noticed that many protein associations affected by IF16 were similarly affected by EOD fasting. Proteins with perturbations in the AL group affected metabolic processes, including carbohydrate and glycogen metabolism, oxidation-reduction, and protein transport (*Figure 3—figure supplement 3* and *Supplementary file 7c*). Among these were phosphofructokinase (PFKL), lactate dehydrogenase A (LDHA), phosphoglycerate dehydrogenase (PHGDH), and kinases AKT2 and DTYMK. Unlike AKT1, which was found to be upregulated by IF, AKT2 did not vary in abundance with any IF regimen. Proteins with association perturbations in IF12 regimen mainly were localized to the mitochondria and showed an inclination towards oxidative stress mechanisms. The IF16 regimen impacted proteins in various cellular processes, including platelet activation, migration, translational control, fatty acid oxidation, and Rap1 signaling. Several association modulations involved integrins, myosin, and actin, including ITGB3 and ITGA2B, which have been linked to cardiac pathologies (*Pillois and Nurden, 2016*). While perturbations accompanying extended EOD fasting maintained some of the functional changes evident in IF16 hearts, such as NADH oxidation and electron transport chain, association alterations also occurred in lipid-related pathways. Importantly, we observed modulation of the PPAR signaling pathway by EOD fasting with key components PDPK1, APOA1, FABP3, PCK2, and SCP2 affected (*Figure 3—figure supplement 3b*). This finding is interesting as PPAR modulators are currently being evaluated as therapeutic players for several cardiac diseases (*Khuchua et al., 2018*).

A consistent pattern across the inferred association perturbations was that most of these involved proteins in the metabolic pathways were also localized to the mitochondria (*Figure 3—figure supplement 4a*). We constructed a mitochondrial association network to identify the modulated associations that occurred in an IF regimen-specific manner. From the network, it was evident that many of the perturbed proteins co-associate, particularly in the EOD group, and share associations relevant to an intricate orchestration of mitochondrial functional protein associations in response to IF (*Figure 3—figure supplement 4b* and *Supplementary file 7d*). Among the mitochondrial proteins affected by IF, some, including UQCRC1, METAP1D, SCP2, COA7, AFG3L2, FAM162A, and COA4, were involved in associations spanning multiple functional processes suggesting coordinating rewiring of various pathways by mitochondrial functions. For instance, UQCRC1 shares IF responsivity with proteins involved in both fatty acid oxidation and carbohydrate metabolism. Overall, our data suggest the involvement of mitochondrial proteins in adaptive responses of the heart to IF.

## IF impacts phosphosignaling pathways in a cardioprotective manner

To gain deeper insight into IF-induced signaling alterations in myocardial cells, we investigated changes in the phosphorylation landscape that accompanies IF (*Figure 4*, *Figure 4—figure supplement 1* and *Supplementary file 8a*). We enriched phosphopeptides from heart tissue homogenates (n=3 from each group) and then profiled them using label-free phosphoproteome analysis. Among the different groups, we identified 12,540 phosphorylated sites on 2931 proteins of which, ~80% and ~19% sites were phosphorylated on serine and threonine respectively, with only 1% being tyrosine phosphorylated. Peptides with single phosphorylation sites represented 81% of the phosphopeptides analyzed. Among the sites identified, 66% corresponded with those curated in the PhosphoSitePlus database (*Hornbeck et al., 2012*). There was high reproducibility of phosphopeptide quantifications among the biological replicates with an average intensity correlation of 0.86.

To elucidate IF-induced changes in protein phosphorylation, we focused on 6935 unique phosphorylation sites from 7615 phosphopeptides that were quantified from at least two biological replicates in the dietary groups (*Supplementary file 8b*). Approximately 84% of these belonged to class I phosphorylation sites with a localization probability ≥0.75, and overall displayed an average localization probability of 0.90, indicative of high phosphosite confidence. On assessing those sites that were significantly altered (Dunnett adjusted p-value <0.05) with IF, we observed that phosphorylation changes varied with increasing duration of the fasting period with EOD exhibiting the greatest number of phosphosite abundance changes, similar to what was observed in the proteome (*Figure 4a* and *Supplementary file 8b*). However, in contrast to proteome remodeling, phosphoproteome profiling revealed increased phosphosite abundance on a majority of the differentially modulated phosphoproteins, and several of these were independent of proteome changes. This is intriguing as it points to differential exploitation of translational and post-translational mechanisms to remodel cardiac cells in response to IF. Site-specific phosphorylation changes occurred on several proteins involved in structural remodeling of the heart, such as titin, which was observed with multiple changes (*Figure 4a*), and on regulatory proteins, including kinases, phosphatases, and enzymes in metabolic pathways (*Figure 4—figure supplement 1*). For instance, phosphorylation changes were observed on several MAPK associated kinases including MAPK1/3 and MAPK12, PKC family kinases including PKC-alpha, -beta, and -theta kinases that coordinate muscle contraction, including MYLK, MYLK3, and PAK2, and sphingolipid signaling kinase SPHK2 (*Supplementary file 8b*). Enzymes involved in lipid metabolism such as ACOT1 and ACSL1, glycolytic enzyme triose phosphate isomerase (TPI1) and fumarate hydratase (FH) in the TCA cycle, phosphorylase kinase (PHKB) involved in glycogen breakdown, all showed IF-responsive changes in phosphosite abundance emphasizing phosphorylation-dependent remodeling of energy homeostasis pathways with IF.

While phosphorylation signatures reflect kinase activities, we noticed that only 2.8% of the phosphosites we quantified had experimentally validated kinases. Hence, we defined kinase-substrate relationships through motif mapping for each IF regimen. Accordingly, we focused on all the modulated phosphosites and identified motifs specifically enriched (p-value <0.05) within distinct phosphosite expression clusters. The clustering revealed discrete patterns of phosphorylation changes that occur in response to each IF regimen (*Figure 4b*). PKA/PKC were commonly activated across all IF regimens. CSNK and p38 signaling were exclusive to the IF16 group, while AMPK, the master regulator of energy metabolism, exhibited activation in longer fasting groups. Overall, the kinase activation profiles were more pronounced in IF16 and EOD groups compared to the AL group. Pathway-based assessment of the modulated phosphoproteins highlighted the conserved modulation of pathways associated with insulin signaling, focal adhesion, and cGMP-PKG signaling across all IF groups (*Figure 4c–d*). cGMP-PKG signaling is known to mediate vasorelaxation, and activation of this pathway is regarded as cardioprotective (*Kukreja et al., 2012*). In line with the kinase prediction, components of AMPK signaling showed significant modulation in both IF16 and EOD groups (*Figure 4c–d*). In summary, our phosphoproteome analyses revealed that IF impacts different signaling pathways that influence the heart's structural and functional plasticity.

To further validate our proteomics data, we carried out immunoblot validation to confirm a few key findings we inferred based on IF-induced proteome and phosphoproteome alterations. For this and subsequent functional studies, we chose the IF16 group as it showed IF-specific rewiring of several kinase-induced pathway rewiring events and is also a more realistic fasting regimen in human. We evaluated the expression levels of selected proteins from various pathways, namely p-AMPKα/

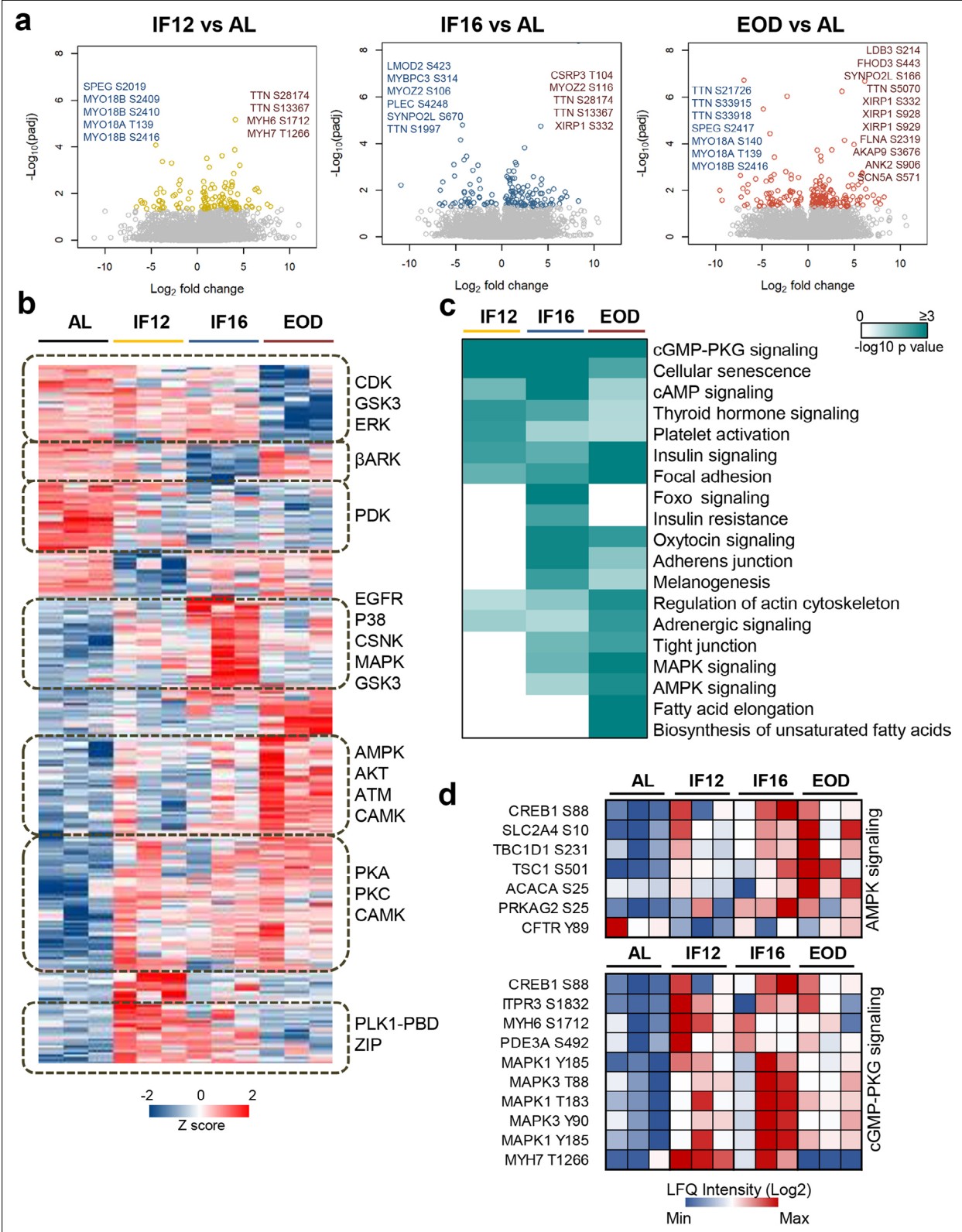

**Figure 4.** Phosphoproteome landscape of IF-induced changes in heart. (**a**) Phosphorylation sites showing significant changes in the hearts of mice in the IF12, IF16, and EOD. Phosphorylation changes observed on structural sarcomere organization proteins are highlighted. (**b**) Hierarchical clustering of differentially altered phosphopeptides showing increased (red) and decreased (blue) phosphosite abundance. Kinase motifs enriched (p<0.05) in representative clusters are shown (Fisher's exact test for enrichment). (**c**) Pathways significantly enriched (p-value ≤0.05) among the differentially altered

*Figure 4 continued on next page*

*Figure 4 continued*

phosphoproteins across the three IF regimens are visualized as a heatmap (Fisher's exact test for enrichment). The color scale represents negative log10-transformed p values. (**d**) Temporal alteration of specific IF-responsive phosphorylation sites in AMPK and cGMP-PKG signaling. n=3 mice in each experimental group.

The online version of this article includes the following figure supplement(s) for figure 4:

**Figure supplement 1.** Site-specific alteration of IF-responsive phosphoproteins.

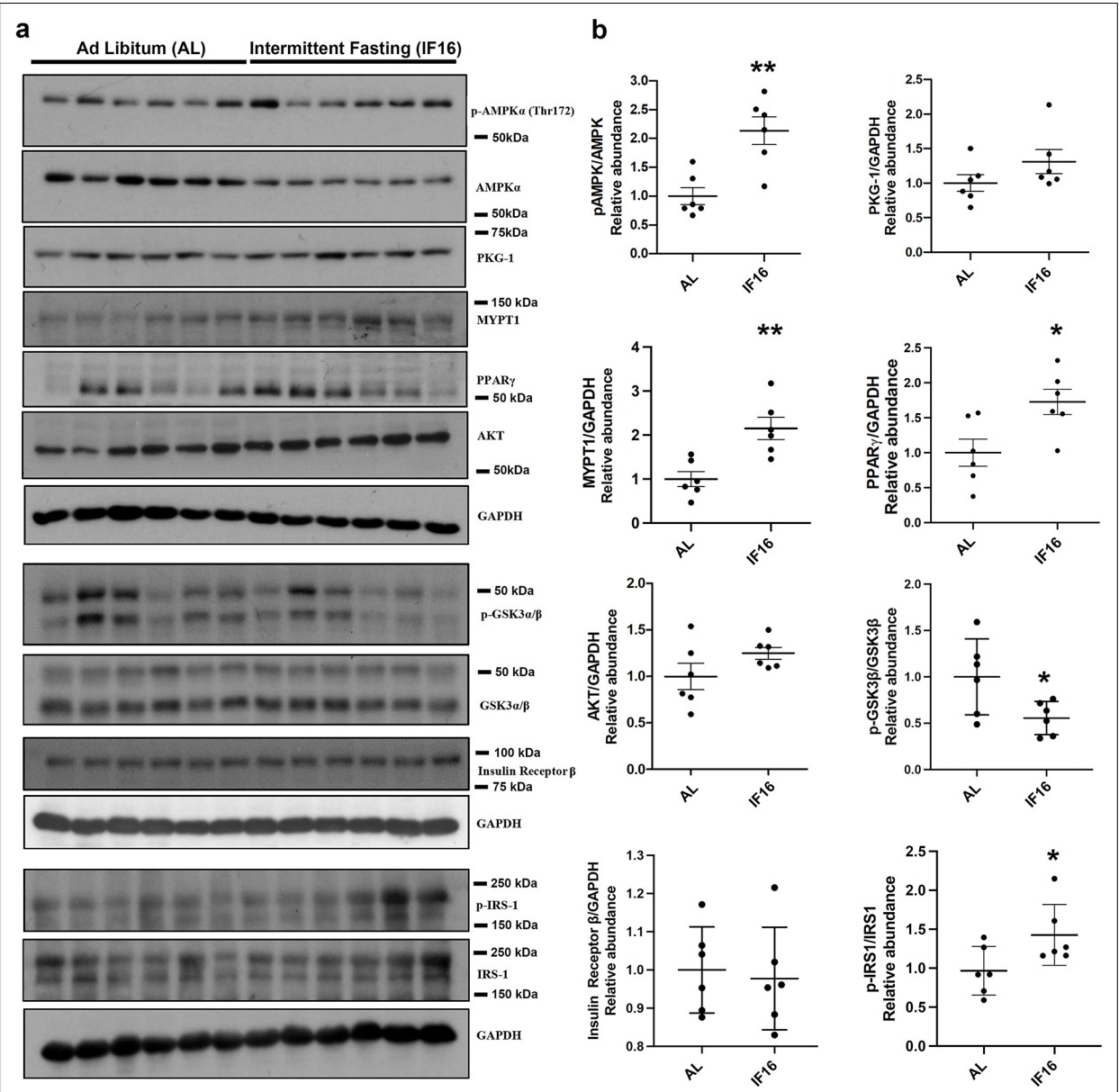

**Figure 5.** Immunoblot validation of the selected proteins in IF-modulated pathways. (**a**) Immunoblots that show the effects of IF16 on the expression of selected proteins p-AMPKα/AMPKα, PKG-1, MYPT1, PPARγ, p-GSKαβ/GSKαβ, AKT, p-IRS1/IRS-1 and Insulin receptor β compared to AL group. Data are represented as mean ± S.E.M. n=6 mice in each experimental group. *p<0.05 compared with young sham; **p<0.01 compared with AL group.

The online version of this article includes the following source data for figure 5:

**Source data 1.** Western blot uncropped membranes.

AMPKα, PKG-1, MYPT1, PPARγ, p-GSKαβ/GSKαβ, AKT, Insulin receptor β, and p-IRS1/IRS-1, in IF16 and AL using immunoblotting (*Figure 5a–b*). In line with our proteomic inferences, compared to the AL group, expression levels of pAMPK/AMPK, PPARγ, MYPT1, and IRS1 were significantly higher in the IF16 group. In addition, we observed increasing trends of PKG-1, AKT and decreasing trend of p-GSKαβ/GSKαβ in the IF16 group compared to the AL group (*Figure 5a–b*). Overall, these results further support the findings of IF-modulated pathways observed from our omics analyses.

## IF improves cardiac performance after stress

With the enhanced cardiac resilience inferred from the omics data, we next investigated the effects of IF on heart function and dynamics in the IF mice compared to the AL mice. For this, we performed echocardiographic measurements in mice that had been maintained on AL or IF16 diets for 6 months. While no differences in heart function were observed between IF and AL groups, AL mice started to show signs of age deterioration, such as prolonged E/A and shortened deceleration time (*de Lucia et al., 2019*; *Figure 6—figure supplement 1*, *Figure 6—videos 1–4*). To evaluate cardiac functional reserve, dobutamine stress tests were performed after basal parameter acquisition. As expected, administration of dobutamine resulted in significant inotropic, lusitropic, and chronotropic cardiac responses *Tyrankiewicz et al., 2013* in both groups. However, the increase in heart rate, as well as ejection fraction, fractional shortening and left ventricular dimensions after stress were significantly higher in IF16 mice compared with AL control mice (*Supplementary file 9*, *Figure 6a*, *Figure 6—videos 1–4*), suggesting that IF preserves myocardial functional reserve. To confirm these findings, strain analyses were performed. Longitudinal and radial strain significantly improved after dobutamine administration in IF16 mice but not in AL mice, suggesting that IF preserved systolic contractile reserve, which deteriorated with age in the AL mice (*Supplementary file 9*, *Figure 6b*).

## Discussion

Increasing evidence from animal and human studies suggests that IF reduces the risk for cardiovascular disease and bolsters myocardial function and resilience (*Mattson et al., 2017*; *Mattson and Wan, 2005*; *Sutton et al., 2018*). However, except for enhancement of parasympathetic tone (*Mager et al., 2006*; *Griffioen et al., 2013*) and upregulation of protein chaperones (*Ma et al., 2019*), the underlying mechanisms are unknown. Collectively, our comprehensive datasets provide a window into the molecular, cellular, structural, and functional remodeling of the heart in mice adapted to IF eating patterns. To address the question of how the duration of the fasting period in specific IF regimens affects the molecular remodeling of the heart, we compared the molecular effects of IF12, IF16, and EOD fasting. All three IF regimens affected the pathways that regulate cellular carbohydrate, lipid and protein metabolism, cell-cell interactions, and myocardial cell contractility. Previous findings suggested that the beneficial effects of IF on organ function and stress resistance require 2–4 weeks to be fully engaged (*Mager et al., 2006*; *Liu et al., 2019*; *Bruce-Keller et al., 1999*; *Johnson et al., 2007*). If and to what extent the molecular remodeling of the heart in response to IF reverts to the pre-IF state upon return to ad libitum feeding remains to be determined.

There are many similarities in the effects of aerobic exercise and IF on the cardiovascular system that beg the question of the extent of overlap in the underlying molecular mechanisms. For example, both IF and regular exercise training increase parasympathetic tone and heart rate variability, and both can protect the myocardium against ischemic injury (*Rengo et al., 2013*; *Voulgari et al., 2013*). Moreover, both exercise and IF stimulate mitochondrial biogenesis in skeletal muscle (*Marosi et al., 2018*; *Drake et al., 2016*). Exercise training also stimulates mitochondrial biogenesis in myocardial cells (*Bernardo et al., 2018*). Studies of the effects of IF on the brain and skeletal muscle suggest that IF activates several major pathways engaged by ischemic and metabolic preconditioning, including upregulation of protein chaperones, anti-apoptotic proteins, and autophagy, and suppression of the mTOR pathway and protein synthesis (*Marosi et al., 2018*; *Mattson and Arumugam, 2018*). Previous research has led to the development of several treatments that mimic one or more effects of exercise, caloric restriction, and IF. Examples include agents that induce mild transient metabolic stress such as 2-deoxyglucose and mitochondrial uncoupling agents (*Wan et al., 2003b*; *Liu et al., 2019*; *Liu et al., 2015*; *Wan et al., 2004*; *Yu and Mattson, 1999*), ketone esters (*Cox et al., 2016*; *Kashiwaya et al., 2013*), and mTOR inhibitors such as rapamycin (*Sciarretta et al., 2014*).

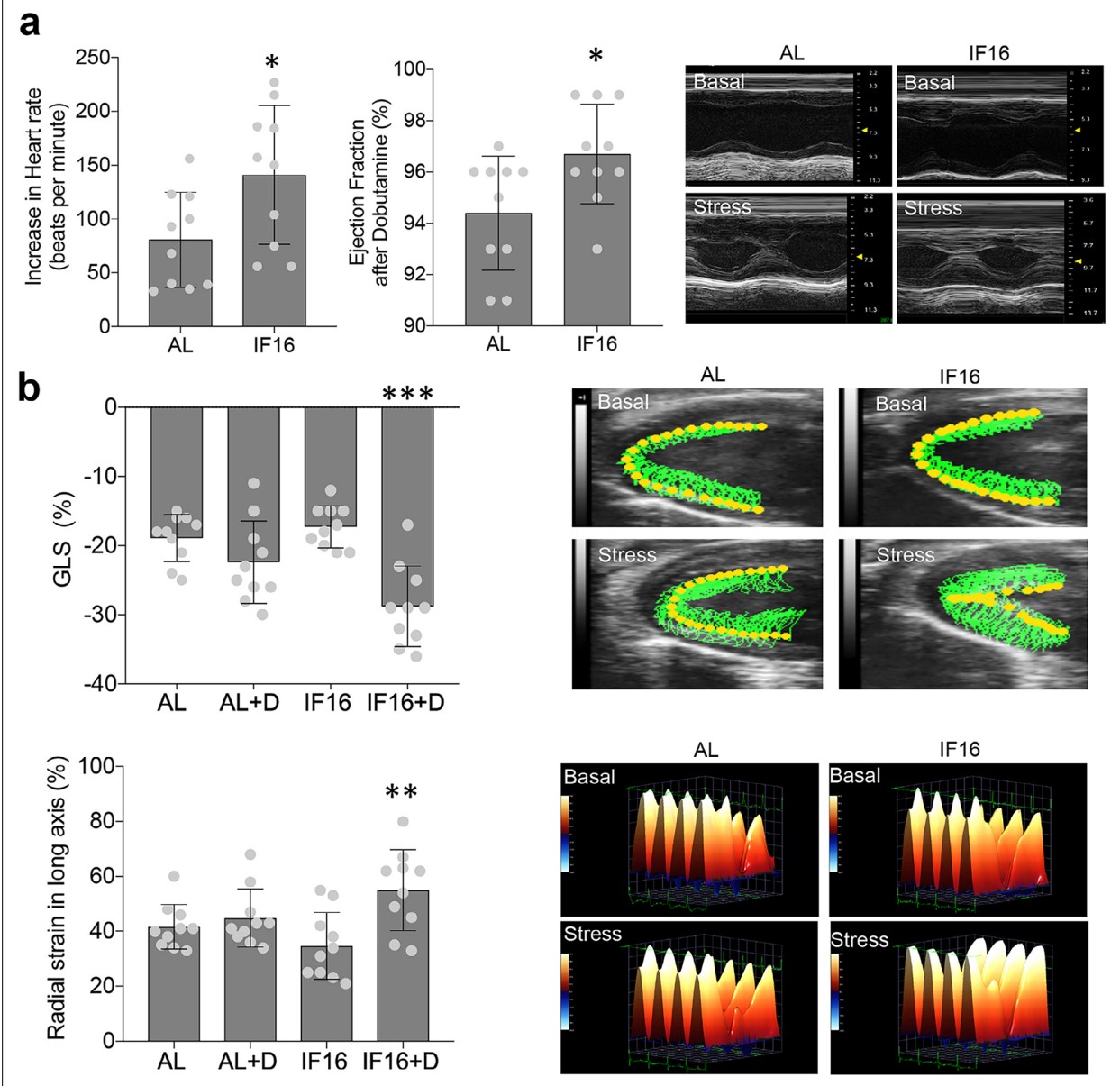

**Figure 6.** Echocardiographic analyses of IF on cardiac stress adaptation. (**a**) Dobutamine stress tests for ability of the heart to adapt to stress under IF. Heart rate and ejection fraction following administration of dobutamine were measured. Values are the mean ± SD; n=10 each; *p<0.05 (Unpaired t-test). Representative images of long axis M-Mode raw data before and after dobutamine-induced stress are shown in the right panel. (**b**) Contractility in hearts of IF16 mice compared with AL mice as illustrated by global longitudinal strain (GLS) (upper panel) and radial strain (lower panel). Values are the mean ± SD (n=10 in each experimental group; **p<0.01, ***p<0.001) before and after dobutamine (one-way ANOVA with Tukey's multiple comparisons test). Representative images of myocardial movement (upper panel) and radial strain (lower panel) before and after dobutamine-induced stress.

The online version of this article includes the following video and figure supplement(s) for figure 6:

**Figure supplement 1.** Echocardiographic analyses.

**Figure 6—video 1.** Baseline ad libitum echocardiographic measurements.
https://elifesciences.org/articles/89214/figures#fig6video1

**Figure 6—video 2.** Baseline *intermittent fasting* echocardiographic measurements.
https://elifesciences.org/articles/89214/figures#fig6video2

**Figure 6—video 3.** Dobutamine stress ad libitum echocardiographic measurements.
https://elifesciences.org/articles/89214/figures#fig6video3

**Figure 6—video 4.** Dobutamine stress *intermittent fasting* echocardiographic measurements.
https://elifesciences.org/articles/89214/figures#fig6video4

Our analysis of functional interactions and IF-induced alterations provides a global view of molecular and metabolic rewiring influencing various aspects of cardiac performance and bioenergetics. We observed IF-induced changes in expression of several enzymes functioning in key energy pathways, as well as kinases involved in cellular signaling. While IF triggered the downregulation of several lipid metabolism enzymes, thioesterases that regulate fatty acid flux, such as ACOT1, ACOT2, and ACOT7, were upregulated, especially in the EOD group. ACOT1, for instance, functions as a molecular rheostat to impede excessive fatty acid oxidation during fasting and dampens associated inflammation and oxidative stress by inducing protective PPARα signaling (*Franklin et al., 2017*). Such an induction of PPARα activity with IF is indeed supported by increased protein abundance of several of its transcriptional targets, including HMGCS2 and PDK4. It is noteworthy that signaling alterations driving such PPAR-associated modulations were also evident from IF-induced phosphorylation changes, including activation of PKA, one of the key modulators of PPAR transcriptional activity, especially in the context of lipid oxidation during fasting and exercise (*Lazennec et al., 2000*). In addition to modulating fatty acid flux, PKA also triggers glycogen breakdown by activating phosphorylase kinase PHKB, which in turn triggers glycogen phosphorylase to release glucose 1 phosphate from glycogen (*Brushia and Walsh, 1999*). Our data point to an increase in phosphorylation site abundance on PHKB with IF suggesting increased turnover of glycogen with IF. The PKA effect of IF was accompanied by changes in associated cAMP signaling which also showed enhanced activation with prolonged durations of fasting as indicated by proteome modulation. The cAMP/PKA axis plays crucial modulatory roles in various aspects of cardiac physiology, including cardiomyocyte contractility via excitation–contraction coupling (ECC), control of heart rate, and downstream relay of β-adrenergic receptor signaling, is known to be perturbed in instances of heart failure (*Bockus and Humphries, 2015*; *Callaghan, 2016*; *Lissandron and Zaccolo, 2006*).

AMPK emerged as another prominent signaling cascade that was activated with longer hours of fasting (IF16 and beyond) based on the modulated phosphorylation sites. We observed that the substrates of AMPK were associated with multiple aspects of energy usage that are known to enhance the metabolic flexibility of the heart. TBC1D1, another substrate that is important in AMPK-induced GLUT4 translocation, also showed a consistent increase in functional phosphorylation with increasing fasting duration (*Jessen et al., 2011*). Disruption of GLUT4 trafficking is associated with impaired glucose uptake and ensuing insulin resistance (*Mueckler, 2001*), and the phosphorylation change on TBC1D1 and GLUT4 along with the predicted increase in AKT activity suggest that IF improves insulin sensitivity in the heart by modulating glucose uptake and resetting aberrations in insulin signaling via increasing phosphorylation of such proteins. The effects of insulin signaling are primarily mediated by AKT. We identified AKT1 to be differentially expressed based on protein abundance, AKT2 to have specific alterations in IF triggered association modulations, and also predicted AKT to be induced based on substrate signatures from our phosphoproteome data. Collectively, our data suggest that AKT may be an important mediator of heart cell responses to IF. In addition, there was a striking increase in phosphorylation of epitopes on numerous proteins known to regulate heart development and cardiomyocyte contraction, particularly that is a crucial determinant of myocardial passive stiffness and whose malfunction is associated with various myopathies (*LeWinter and Granzier, 2014*; *Makarenko et al., 2004*). In all, these highlight varying degrees of kinase and metabolic rewiring in the heart induced by different durations of IF, with the most protective effects accompanying prolonged fasting.

Whilst our data provide a comprehensive molecular snapshot of the changes accompanying various courses of IF, there is a limitation that our landscape is reflective of net changes occurring across the entire heart. Thus, it is challenging to understand specific changes within the different cell types that comprise the heart. Such a limitation is exaggerated, particularly for the transcriptome analysis, as cardiomyocytes account for only 30% of cells in the mammalian heart and yet account for the majority of the cardiac mass (*Zhou and Pu, 2016*). Knowledge of cell type-specific responses to IF will therefore be important to delineate distinct responses of heart cells to various IF regimens, and future studies exploring single-cell sequencing will add new dimensions to our understanding of cardiac dynamics with IF.

Further experimentation with different aspects of our study is required for detailed understanding of our findings. The timing of fasting cycles in this study may not promote synchronization of the natural circadian rhythm of nocturnal rodents, which may impact the potential for translating these to

other model systems and understanding human patho-physiology. It has been established that time-restricted feeding/intermittent fasting cycles impact and can reset secondary tissue-specific peripheral clocks independent of the master circadian clock that is generally phase-locked to light-dark cycles (*Damiola et al., 2000*; *Hara et al., 2001*; *Stokkan et al., 2001*). While such mechanisms have been well documented in the context of liver and hepatic clocks, the impact of feeding on cardiac clock rhythmicity, however, remains to be deciphered. Evidence from our data suggests that fasting regimens affect circadian oscillations in heart cells, at least in part, independently of the central light-dark cycle-responsive clock. In addition, our analysis of cardiac function only included the AL and IF16 groups. We selected the IF16 fasting regimen to perform proof of principle functional experiments to further understand the functional outcome of proteomic changes observed. Echocardiographic analysis of the EOD group may provide significant information on the effect of IF, as the EOD group had the largest effect sizes from multi-omics analyses. With our study highlighting several kinase rewiring and signaling pathways impacted by IF based on modulations in proteome and phosphorylation site modulations, it will be interesting to follow-up on several of these to further assess their functional significance in the context of heart health and function. It is noteworthy that previous studies established that the EOD regimen leads to hearts with thinner walls and a lower left ventricular end-diastolic volume (LVEDV) (*Ahmet et al., 2005*) and lower blood pressure and heart rate (*Wan et al., 2003a*) than those of control animals. The latter effects of EOD were associated with cardioprotection in a rat model of myocardial infarction (*Ahmet et al., 2005*). Furthermore, our multi-omics study was limited to mice 6–9 months old, and it is, therefore, difficult to draw conclusions as to the effects of IF on ageing heart functions. All three IF groups maintained similar average body weights throughout the study period and were significantly lower than the AL group. Therefore, the different transcriptomes, proteomes, and phosphoproteomics observed among the three IF groups were independent of body weight. Our data suggest overlapping but distinct molecular adaptations of the heart to different intermittent fasting regimens.

In conclusion, our comprehensive multi-omics analysis sheds light on the intricate molecular mechanisms underlying the beneficial effects of intermittent fasting (IF) on cardiac health and disease prevention. Our findings demonstrate that various IF regimens induce distinct and time-dependent changes in the cardiac proteome, with longer fasting periods resulting in more significant alterations. The EOD group, in particular, exhibited the most significant protein abundance changes. The unsupervised hierarchical clustering analysis of the altered heart proteome highlights unique protein expression patterns and co-expressed protein clusters in all IF groups compared to the AL group. Collectively, these results provide valuable insights into potential pharmacological targets for developing novel therapeutic strategies for cardiovascular diseases. Future studies may further explore these targets and the underlying pathways to advance the development of effective interventions to improve cardiac health.

## Acknowledgements

*Supplementary file 1a* in this article was created using BioRender. Funding This work was supported by ODPRT, National University of Singapore, National Medical Research Council Research Grants (NMRC-CBRG-0102/2016 and NMRC/OFIRG/0036/2017), Singapore and Agency for Science, Technology and Research, Singapore and the start up fund from La Trobe University, Melbourne, Australia.

## Additional information

### Funding

| Funder | Grant reference number | Author |
| --- | --- | --- |
| National Medical Research Council | NMRC-CBRG-0102/2016 | Thiruma V Arumugam |

| Funder | Grant reference number | Author |
| --- | --- | --- |
| National Medical Research Council | NMRC/OFIRG/0036/2017 | Thiruma V Arumugam |
| Agency for Science, Technology and Research | | Jayantha Gunaratne |
| La Trobe University | | Thiruma V Arumugam |

The funders had no role in study design, data collection and interpretation, or the decision to submit the work for publication.

### Author contributions

Thiruma V Arumugam, Conceptualization, Resources, Formal analysis, Supervision, Validation, Investigation, Methodology, Writing – original draft, Project administration, Writing – review and editing; Asfa Alli-Shaik, Data curation, Software, Formal analysis, Validation, Investigation, Visualization, Methodology, Writing – original draft, Writing – review and editing; Elisa A Liehn, Conceptualization, Resources, Formal analysis, Investigation, Methodology, Writing – review and editing; Sharmelee Selvaraji, Validation, Investigation, Writing – review and editing; Luting Poh, Vismitha Rajeev, David Tan Zhi Hao, Chutima Rattanasopa, David Castano Mayan, Gavin Yong-Quan Ng, Sang-Ha Baik, Investigation; Yoonsuk Cho, Yongeun Cho, Jongho Kim, Investigation, Methodology; Joonki Kim, Investigation, Visualization, Methodology; Hannah LF Swa, Validation, Investigation, Methodology; David Yang-Wei Fann, Investigation, Methodology, Writing – review and editing; Karthik Mallilankaraman, Resources, Writing – review and editing; Mathias Gelderblom, Conceptualization, Writing – review and editing; Grant R Drummond, Christopher G Sobey, Writing – review and editing; Brian K Kennedy, Conceptualization, Resources, Writing – review and editing; Roshni R Singaraja, Conceptualization, Resources, Supervision, Writing – review and editing; Mark P Mattson, Conceptualization, Writing – original draft, Writing – review and editing; Dong-Gyu Jo, Conceptualization, Resources, Supervision, Writing – original draft, Writing – review and editing; Jayantha Gunaratne, Conceptualization, Resources, Software, Formal analysis, Supervision, Funding acquisition, Investigation, Methodology, Writing – original draft, Project administration, Writing – review and editing

### Author ORCIDs

Thiruma V Arumugam ⓘ https://orcid.org/0000-0002-3377-0939
Asfa Alli-Shaik ⓘ https://orcid.org/0000-0001-7477-0495
Karthik Mallilankaraman ⓘ http://orcid.org/0000-0002-9492-9050
Christopher G Sobey ⓘ http://orcid.org/0000-0001-6525-9097
Roshni R Singaraja ⓘ http://orcid.org/0000-0002-3418-3867
Jayantha Gunaratne ⓘ http://orcid.org/0000-0002-5377-6537

### Ethics

All in vivo experimental procedures were approved by the National University of Singapore (Ethics approval number: R15-1568) and La Trobe University (Ethics approval number: AEC21012) Animal Care and Use Committees and performed according to the guidelines set forth by the National Advisory Committee for Laboratory Animal Research (NACLAR), Singapore, and Australian Code for the Care and Use of Animals for Scientific Purposes (8th edition) and confirmed NIH Guide for the Care and Use of Laboratory Animals.

Reviewer #1 (Public Review): https://doi.org/10.7554/eLife.89214.2.sa1
Reviewer #2 (Public Review): https://doi.org/10.7554/eLife.89214.2.sa2
Author Response https://doi.org/10.7554/eLife.89214.2.sa3

# Additional files

### Supplementary files

- Supplementary file 1. List of all identified proteins in heart tissues of different dietary groups.

- Supplementary file 2. IF-responsive differentially expressed proteins in heart tissues among different dietary groups.

- Supplementary file 3. Functional processes and pathways enriched among different IF groups.
- Supplementary file 4. RNA-seq based transcriptome profiling for heart tissues obtained from mice in different IF regimens.
- Supplementary file 5. Functional processes and pathways differentially enriched in proteome and transcriptome.
- Supplementary file 6. Node properties of time-resolved IF protein network.
- Supplementary file 7. Perturbation of protein association co-regulation in different dietary groups.
- Supplementary file 8. Quantitative phosphoproteome of heart tissues across different dietary groups.
- Supplementary file 9. Echocardiographic assessment of myocardial functional and cardiac dynamics.
- MDAR checklist

## Data availability

The mass spectrometry proteomics data have been deposited to the ProteomeXchange Consortium via the PRIDE partner repository with the dataset identifier PXD015452. High throughputRNA sequencing data from this manuscript have been submitted to the NCBI Sequence Read Archive (SRA) under accession number GSE133290.

The following datasets were generated:

| Author(s) | Year | Dataset title | Dataset URL | Database and Identifier |
|---|---|---|---|---|
| Alli-Shaik A, Gunaratne J | 2023 | Multiomics Analyses Reveal Dynamic Bioenergetic Pathways and Functional Remodeling of the Heart During Intermittent Fasting | https://www.ebi.ac.uk/pride/archive/projects/PXD015452 | PRIDE, PXD015452 |
| Arumugam TV, Kim J | 2023 | Transcriptome analysis of mouse heart under different intermittent fasting conditions | https://www.ncbi.nlm.nih.gov/geo/GSE133290 | NCBI Gene Expression Omnibus, GSE133290 |

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
