## [Editor Report · eLife assessment]

This study provides a **useful** catalog of the cardiac proteome and transcriptome in response to intermittent fasting. Although mechanistic integration is limited, the technical aspects have been executed in a **solid** way, and sufficient evidence is provided to support the main conclusions. Future work can build on this study to expand our understanding of the relationship between dietary perturbations and cardiac function.

---

## [Referee Report · Reviewer #1 (Public Review)]

In this manuscript, the authors explored the benefits of intermittent fasting on the cardiac physiology through a multi-omics approach and compared different fasting times (IF12; IF 16 and EOD) for a duration of 6 months. Combining the RNA-sequencing, proteomics and phosphor-proteomics analysis, the authors have made an interesting observation that different fasting times would lead to different changes that could be important for the cardiac physiology. Moreover, the changes observed at transcriptional level are different from protein level, suggesting a post-transcriptional regulation mechanism. Using western blot, the authors have confirmed the key signaling pathways, including AMPK, IRS pathway to be significantly altered upon intermittent fasting for 16hrs. Lastly, as a proof of concept for better cardiac function, the animals were challenged with dobutamine and echocardiography was performed to show the mice subjected to intermittent fasting have better cardiac systolic function.

The impact of intermittent fasting on cardiovascular health has been well characterized in several studies. This report appears to be the first one utilizing a multi-omics approach and provided an interesting dataset at transcriptome, proteome and phosphor-proteome levels, and would serve as a valuable data resource for the field. I have the following concerns:

Major concerns:

1. The rationale for choosing the intermittent fasting pattern and timing

2. Lack of validation study

3. Poor western blot image quality

---

## [Referee Report · Reviewer #2 (Public Review)]

This study provides an unbiased characterization of the cardiac proteome in the setting of intermittent fasting. The findings constitute a resource of quantitative proteomic data that sheds light on changes in cardiac function due to diet and that may be used in the future by other investigators. There are a number of key missing details that limit interpretation or present opportunities to strengthen the study. For example, the authors find that apolipoproteins are altered with fasting but it is not clear whether this is a contribution of myocardial tissue changes or systemic effects spilling into blood in cardiac tissues. Some statements in the text like "Approximately one-third of the differentially expressed proteins in IF groups compared to AL were enzymes with catalytic activity involved in energy homeostasis pathways" do not appear to be supported by data. It is not clear how the list of Kinases were generated for Figure 1B. Changes in chromatin or gene expression are not measured so the conclusion that EOD led to 'epigenetic changes' relative to IF16 is not well supported. There are also a number of areas where the text is vague. For example, it is not clear what is meant by 'trend shift' when discussing EOD results and Figure 3 generally could use additional information to better understands the figures. An interesting finding is that the IF16 groups showed cardiac hypertrophy (SFig 11b). This is potentially a novel finding and the text should elaborate more on this phenomenon.

---

## [Author Response]

**Reviewer #1 (Public Review):**
In this manuscript, the authors explored the benefits of intermittent fasting on the cardiac physiology through a multi-omics approach and compared different fasting times (IF12; IF 16 and EOD) for a duration of 6 months. Combining the RNA-sequencing, proteomics and phosphor-proteomics analysis, the authors have made an interesting observation that different fasting times would lead to different changes that could be important for the cardiac physiology. Moreover, the changes observed at transcriptional level are different from protein level, suggesting a post-transcriptional regulation mechanism. Using western blot, the authors have confirmed the key signaling pathways, including AMPK, IRS pathway to be significantly altered upon intermittent fasting for 16hrs. Lastly, as a proof of concept for better cardiac function, the animals were challenged with dobutamine and echocardiography was performed to show the mice subjected to intermittent fasting have better cardiac systolic function.The impact of intermittent fasting on cardiovascular health has been well characterized in several studies. This report appears to be the first one utilizing a multi-omics approach and provided an interesting dataset at transcriptome, proteome and phosphor-proteome levels, and would serve as a valuable data resource for the field. I have the following concerns:Major concerns:1. The rationale for choosing the intermittent fasting pattern and timingWhile the 16:8 intermittent fasting is relatively standard, what is the rationale to test IF 12 hours? As a 4-hour fasting difference might not cause dramatic changes in transcriptome and proteome. Also, what is the rationale to perform 6 months study? The dobutamine stress test is not a terminal procedure, have the authors examined the cardiac function prior to 6 months to see whether there is a difference?

We sincerely thank the reviewer for providing insightful comments and feedback on our study. The aim of our research is to gain a comprehensive understanding of molecular reprogramming in the heart during intermittent fasting using multi-omics techniques. We acknowledge the reviewer's concern regarding the selection of three different time points for intermittent fasting. Our rationale for choosing these time points was to align with the practices commonly used by researchers in the field. By doing so, we intended to explore and compare the effects of different intermittent fasting regimens on the heart. Through our study, we found that a longer fasting period resulted in the most significant changes in the proteome abundance. Though we agree that 4-hour fasting difference may not significantly alter transcriptome and proteome in terms of expressions, remarkable changes of post-translational modifications such as phosphorylation can occur during shorter time periods and this is evident based on the analyses of the modulated phosphoproteins. Hence, we included 12 hours time point also to our analysis. In fact, we would like to emphasize that all three fasting regimens had notable effects on pathways regulating cellular carbohydrates, lipid and protein metabolism, cell-cell interactions, and myocardial cell contractility. Regarding the duration of our study, we opted for a 6-month duration of intermittent fasting to investigate the impact of chronic intermittent fasting on heart transcriptome and proteome changes. While shorter-term (2-3 months) intermittent fasting studies in animals also have shown beneficial effects, we wanted to delve deeper into the molecular alterations induced by long-term intermittent fasting. We acknowledge the reviewer's observation about the dobutamine stress test not being a terminal procedure. In our manuscript, we aimed to present extensive resource data offering molecular insights into intermittent fasting-induced structural and signaling changes in the heart, focusing on various intermittent fasting time intervals. Additionally, we included the effect of cardiac function in response to intermittent fasting, specifically examining the intermittent fasting 16 hours (IF16) group, and highlighted key pathway modulations at this time point as supporting evidence. We appreciate the reviewer’s concern about examining cardiac function prior to 6-month. Although we did not perform this analysis in the current study, we fully agree that such comparison is required for a better understating of the temporal effects of molecular pathways in relation to heart functions during the course of intermittent fasting.

1. Lack of validation study. One interesting observation from this study is the changes of transcriptome does not reflect all the changes at protein level and there is a differential gene expression pattern in IF12, IF16 and EOD. If this is the case, the authors should select a few important targets and provide both mRNA and protein level analysis, as a proof of concept for the bioinformatics analysis accuracy.

We appreciate the reviewer's attention to the comparison of proteome and transcriptome data across different intermittent fasting regimens, as well as their interest in understanding any specific deviations in dietary regimens or sets of proteins. Indeed, it is well-established that post-transcriptional regulation can lead to discrepancies between mRNA and protein levels, primarily due to translational control or protein degradation mechanisms. Posttranscriptional buffering of proteins, particularly enzymes and kinases, is a plausible explanation, given their regulation through post-translational modifications, such as phosphorylations or allosteric regulations. Despite observing a modest correlation between the proteome and transcriptome data, which is generally common, we did identify certain enzymes, such as HMGC2, PDK4ACOT, CLPX, and RNASE4, with a high level of concordance between protein and mRNA abundances. These instances of agreement between the two data types suggest a coordinated regulation of these enzymes at the transcriptional and translational levels during intermittent fasting. To facilitate a clearer understanding of the correlation between proteome and transcriptome data, we have included correlation levels next to the scatter plots in our manuscript. These annotations aim to provide additional insights and aid readers in assessing the relationship between the two datasets.

1. Poor western blot image quality. The quality of the western blot has several issues: a. the change of pAMPK/AMPK appears to be a decrease of total AMPK instead of change at p-AMPK level. Same with GSK3a/b. There appears to be an increase of total GSK3a/b. The AKT should also be blotted and quantified at phosphorylation level. The western blot should be clearly labeled, for the ones with double bands, including GSK3a/b, the author should clearly label which is GSK3a and which is GSK3b. For the IRS with non-specific band, the author should point out IRS-1 band itself.

We appreciate the reviewer's careful evaluation of our study and acknowledge the concerns raised regarding the quality of the western blot images. Despite revising these experiments multiple times, we acknowledge that the immunoblot images may not meet the highest quality standards. We have included the original immunoblots in the supplementary section to ensure transparency and provide additional data for reference.

**Reviewer #2 (Public Review):**
This study provides an unbiased characterization of the cardiac proteome in the setting of intermittent fasting. The findings constitute a resource of quantitative proteomic data that sheds light on changes in cardiac function due to diet and that may be used in the future by other investigators. There are a number of key missing details that limit interpretation or present opportunities to strengthen the study.1. For example, the authors find that apolipoproteins are altered with fasting but it is not clear whether this is a contribution of myocardial tissue changes or systemic effects spilling into blood in cardiac tissues.

We appreciate the reviewer's consideration of the potential effect of spilling blood on our study results. While we agree that such an effect is possible, we would like to emphasize that the observed overall changes in the proteome profile, particularly in pathways regulating metabolism and other cardiac remodeling-associated processes, suggest that the alterations we observed are more likely attributed to changes within the myocardial tissues themselves. We would like to highlight that blood microparticles or extracellular proteins were not enriched in our proteome data and hence the impact of blood spilling is not a concern. In fact, the biological processes we observed were majorly associated with ECM receptor interaction, focal adhesion and signaling pathways, which are not typical for secreted or extracellular proteome encompassing blood leakage.

1. Some statements in the text like "Approximately one-third of the differentially expressed proteins in IF groups compared to AL were enzymes with catalytic activity involved in energy homeostasis pathways" do not appear to be supported by data.

The enzymes among all the differentially expressed proteins in the intermittent fasting (IF) groups compared to the ad libitum (AL) control group are indicated in Supplementary Table S2. This constitutes one-third of the total number of differentially expressed proteins and several of these are involved in metabolic and energy homeostasis pathways.

1. It is not clear how the list of Kinases were generated for Figure 1B.

For the kinases indicated in Figure 1B, all the kinases from the proteins that were differentially expressed among the different dietary regimens compared to the control ad libitum (AL) group were first identified (listed in Supplementary Table S2), followed by enrichment analysis (FDR ≤ 0.05) of the identified kinases across different pathways identified from KEGG pathways derived from DAVID bioinformatics resources.

1. Changes in chromatin or gene expression are not measured so the conclusion that EOD led to 'epigenetic changes' relative to IF16 is not well supported.

We appreciate the reviewer's feedback. Our statement in the manuscript referred specifically to the changes observed in Figure 2, where we presented increased proteomic abundance in pathways related to chromatin remodeling, chromatin organization, gene expression regulation, and histone modification in the EOD (Every Other Day Fasting) group compared to the IF 16 (Intermittent Fasting for 16 hours) group based on functional process and pathway enrichment analysis. Our comprehensive bioinformatics analysis, depicted in Figure 2, provides intriguing insights into these pathways. We acknowledge that further validation and in-depth studies through additional experiments and functional assays are essential to strengthen the conclusion from such observations, which is beyond the scope of the current study. We thank the reviewer for such valuable suggestions that are very useful for our ongoing studies, where we aim to obtain a more robust and thorough understanding of the impact of intermittent fasting on chromatin-related processes.

1. There are also a number of areas where the text is vague. For example, it is not clear what is meant by 'trend shift' when discussing EOD results and Figure 3 generally could use additional information to better understands the figures.

We would like to clarify that the term 'trend shift' refers to the change in the direction of protein and transcript level alterations. Based on the 2-D enrichment analyses that revealed correlated and non-correlated functional processes at the proteome and transcriptome levels, it was evident that during the early intermittent fasting 12 hours (IF12) regimen, the abundance changes of the proteins and transcripts involved in these processes were altered in the same direction (Supplementary Fig. 4b). Nevertheless, with increased fasting hours, mainly in the Every Other Day Fasting (EOD) group, we observed that the levels of proteins and transcripts involved in several of the functional processes appeared to be non-correlated as compared to the IF12 group (Fig. 2d). In Figure 3, we summarize the overall altered protein networks associated with the different intermittent fasting regimens, highlighting densely connected clusters of proteins along with their associated biological processes and pathways. Additionally, we unravel the impact of intermittent fasting on transcriptional rewiring and highlight regimen-specific alterations of specific transcriptional factors, several of which were found to have metabolic implications.

1. An interesting finding is that the IF16 groups showed cardiac hypertrophy (SFig 11b). This is potentially a novel finding and the text should elaborate more on this phenomenon.

We sincerely thank the reviewer for bringing attention to this intriguing aspect of our study. The data you have highlighted warrants further investigation, and we are committed to delving deeper into this area in our future research.